# Holistic Continual Learning under Concept Drift with Adaptive Memory Realignment

**Alif Ashrafee**                                                               *aa5264@rit.edu*
*Rochester Institute of Technology, New York, USA*

**Jędrzej Kozal**                                                      *jedrzej.kozal@pwr.edu.pl*
*Wroclaw University of Science and Technology, Wroclaw, Poland*

**Michał Woźniak**                                                   *michal.wozniak@pwr.edu.pl*
*Wroclaw University of Science and Technology, Wroclaw, Poland*

**Bartosz Krawczyk**                                                 *bartosz.krawczyk@rit.edu*
*Rochester Institute of Technology, New York, USA*

**Reviewed on OpenReview:** *https://openreview.net/forum?id=1drDltOCLM*

## Abstract

Traditional continual learning methods prioritize knowledge retention and focus primarily on mitigating catastrophic forgetting, implicitly assuming that the data distribution of previously learned tasks remains static. This overlooks the dynamic nature of real-world data streams, where concept drift permanently alters previously seen data and demands both stability and rapid adaptation. We introduce a holistic framework for continual learning under concept drift that simulates realistic scenarios by evolving task distributions. As a baseline, we consider Full Relearning (FR), in which the model is retrained from scratch on newly labeled samples from the drifted distribution. While effective, this approach incurs substantial annotation and computational overhead. To address these limitations, we propose Adaptive Memory Realignment (AMR), a lightweight alternative that equips rehearsal-based learners with a drift-aware adaptation mechanism. AMR selectively removes outdated samples of drifted classes from the replay buffer and repopulates it with a small number of up-to-date instances, effectively realigning memory with the new distribution. This targeted resampling matches the performance of FR while reducing the need for labeled data and computation by orders of magnitude. To enable reproducible evaluation, we introduce four concept drift variants of standard vision benchmarks: Fashion-MNIST-CD, CIFAR10-CD, CIFAR100-CD, and Tiny-ImageNet-CD, where previously seen classes reappear with shifted representations. Comprehensive experiments on these datasets using several rehearsal-based baselines show that AMR consistently counters concept drift, maintaining high accuracy with minimal overhead. These results position AMR as a scalable solution that reconciles stability and plasticity in non-stationary continual learning environments. Full implementation of our framework and concept drift benchmark datasets are available at: https://github.com/AlifAshrafee/CL-Under-Concept-Drift.

## 1 Introduction

In recent years, there has been a lot of progress (Masana et al., 2020; Kirkpatrick et al., 2016; Chaudhry et al., 2018; Buzzega et al., 2020a; Arani et al., 2022; Zhuo et al., 2023) in Continual Learning (Chen & Liu, 2018) research. The key challenge in this field is to ensure that models can learn over time while retaining information from earlier tasks and mitigating catastrophic forgetting (French, 1999) — a phenomenon where previously learned knowledge is overwritten by new information. Despite this progress, much of the existing

work assumes that the properties of past data remain static (Masana et al., 2020) once learned. While simplifying the problem, such an assumption overlooks the dynamic nature of real-world data streams (Gama et al., 2014a), where concept drift—shifts in the statistical properties of previously seen data—is a frequent occurrence.

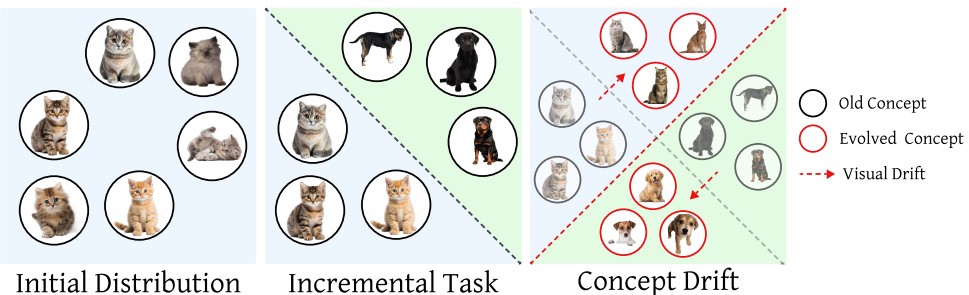

Figure 1: Visualization of concept drift in continual learning. (a) Initial Distribution: The learning process begins with a class of kittens. (b) Incremental Task: A new task introduces adult dogs, prompting the model to form a decision boundary that separates kittens from dogs. (c) Concept Drift: Over time, kittens evolve into adult cats, and adult dogs are replaced by puppies. Although the class labels remain the same (cats vs. dogs), their visual representation shifts, requiring an update in the decision boundary to maintain correct classification.

Concept drift (Widmer & Kubat, 1993) poses a unique challenge for continual learning, as it requires models not only to retain knowledge but also to adapt to changes in previously encountered classes (see Figure 1). In traditional continual learning, the changes in distributions of already learned classes are considered in the Domain-Incremental scenario (van de Ven & Tolias, 2019), where no new classes are introduced over time. On the other hand, the Class-Incremental scenario (van de Ven & Tolias, 2019; Korycki & Krawczyk, 2024) assumes that changes in data distribution occur only by introducing completely new classes, with no shifts in previously learned ones. However, the combination of both scenarios (Korycki & Krawczyk, 2021; Lyu et al., 2024) is rarely addressed in the literature. Recently introduced Class-Incremental Learning with repetition scenario (Hemati et al., 2023) assumes that past classes can reappear in the future, but with distribution of class remaining the same. Considering changes in both past and current class distributions could lead to development of more flexible algorithms, that could handle complexity more easily.

Among the various strategies developed for continual learning, rehearsal methods (Buzzega et al., 2020a; Caccia et al., 2022; Zhuo et al., 2023; Arani et al., 2022) have achieved remarkable success. These methods maintain a small memory buffer of past data samples (Chaudhry et al., 2019), which are replayed during training to mitigate catastrophic forgetting. Korycki & Krawczyk (2021) adapted rehearsal algorithms for continual learning under concept drift by using class centroids to determine whether past representations should be relabeled as a new class. While their approach improves performance, it has several limitations. First, it overlooks the widely adopted reservoir sampling algorithms (Vitter, 1985), which are now standard in most rehearsal-based methods. Second, the assumption that past representations can be reused as new classes conflicts with certain types of concept drift. In cases of domain shift, where class feature distributions change, newly sampled examples may have entirely different representations. Relabeling old samples after drift detection may not enhance classification accuracy and could unnecessarily occupy buffer memory.

**Data Streams and Continual Learning - two sides of the same coin:** Concept drift and evolving data distributions expose the complementary strengths and limitations of continual learning and data-stream mining. Although these two fields have historically developed along parallel trajectories, they address fundamentally intertwined aspects of learning in dynamic environments. Whereas continual learning stresses knowledge retention, safeguarding past information against catastrophic forgetting, data-stream mining stresses knowledge adaptation, enabling models to respond quickly and accurately to evolving data distributions and concept drift. A method that focuses solely on adaptation may achieve only locally optimal performance by discarding valuable long-term knowledge, whereas one that clings rigidly to prior knowledge risks retaining outdated or irrelevant information. Unifying these perspectives is therefore essential for de-

veloping learning systems that are both resilient and adaptive. By bridging insights from both fields, we aim to advance toward holistic approaches that can simultaneously remember and evolve.

**Main contributions:** To the best of our knowledge, we present the first continual-learning framework that explicitly accounts for representation-level concept drift. Our solution couples a lightweight drift-detection module with Adaptive Memory Realignment (AMR): a drift-aware buffer-update strategy that preserves relevant past knowledge while rapidly adapting to evolving distributions. By addressing representation shift directly, the framework models real-world non-stationary streams more realistically and integrates seamlessly with existing continual-learning architectures. Our contributions are:

- **A framework for continual learning under concept drift.** We design a framework for continual learning that enables the simulation of diverse concept drift scenarios across multiple benchmark datasets and severity levels through a set of configurable parameters.

- **Concept-drift-adaptive memory.** We propose AMR, a buffer-update mechanism that mitigates gradient misalignment by selectively removing outdated samples while maintaining robustness to catastrophic forgetting.

- **Efficiency in data and compute.** Extensive experiments on Fashion-MNIST, CIFAR10, CIFAR100, and Tiny-ImageNet show that AMR recovers accuracy after drift with minimal labeled data and low computational overhead.

Together, these contributions provide a scalable, holistic solution that reconciles stability and plasticity for continual learning in truly non-stationary environments.

## 2 Related Work

**Continual Learning:** Efforts to address catastrophic forgetting can be broadly categorized into three approaches (Masana et al., 2020): rehearsal methods, regularization techniques, and network expansion strategies.

Regularization methods constrain the learning process to reduce forgetting. Kirkpatrick et al. (2016) proposed a regularization approach using the Fisher Matrix to preserve key parameters from previous tasks. Similarly, Zenke et al. (2017) introduced a per-parameter regularization technique based on quadratic loss. Another method, presented in Li & Hoiem (2016), use predictions from a model trained on prior tasks as a knowledge distillation-based regularizer for new tasks. Petit et al. (2023) developed a pseudo-feature generation strategy that freezes the backbone after the initial task. In the work of Zhuang et al. (2024), a frozen backbone is also utilized, but the authors frame Continual Learning as a Concatenated Recursive Least Squares problem to compute weight updates through a closed-form solution.

Rehearsal methods mitigate forgetting by retrieving data from past tasks using memory buffers. Even introducing a few learning examples from past tasks (Chaudhry et al., 2019) could limit forgetting. More advanced rehearsal methods incorporate knowledge distillation (Buzzega et al., 2020a), asymmetrical cross-entropy loss (Caccia et al., 2022) or gradient projection (Chaudhry et al., 2018). Some methods use data from the buffer to reduce the recency bias in the classifier layer (Wu et al., 2019). Other methods specify what samples should be selected for storage (Buzzega et al., 2020b; Aljundi et al., 2019) or how to effectively retrieve samples from the buffer (Harun et al., 2024). Recently, weight interpolation between old and new networks was proposed as a complementary mechanism to experience replay Kozal et al. (2024).

Architecture-based algorithms (Rusu et al., 2016) address catastrophic forgetting by expanding network capacity. Typically, they mitigate forgetting by freezing parameters associated with previous tasks (Rusu et al., 2016). However, adapting to concept drift can be challenging with such approaches, and as a result, this category will not be the focus of our work.

**Adaptive and plastic Continual Learning:** Recent continual learning research increasingly targets adaptive plasticity as a first-class objective beyond mitigating forgetting, by explicitly decoupling stable and plastic components in the learning dynamics and parameterization. For instance, Liang & Li (2023) proposes loss

decoupling to separate objectives that govern new-versus-old discrimination and new class learning to better control the stability-plasticity trade-off in task-agnostic CL. Prompt-based designs similarly split functionality into modules specialized for retention and rapid acquisition. PromptFusion (Chen et al., 2024) introduces dedicated stabilizer/booster prompts to disentangle forgetting mitigation from learning new tasks. In the foundation-model regime, SD-LoRA (Wu et al., 2025) improves scalability by decoupling the magnitude and direction of low-rank updates, enabling strong stability-plasticity behavior without rehearsal. Complementing architectural/parameter decoupling, Self-Normalized Resets (Farias & Jozefiak, 2025) directly combats plasticity loss by selectively resetting neurons deemed inactive, restoring the model's ability to adapt late in long task streams. Finally, principled combination strategies such as BECAME (Li et al., 2025) use Bayesian formulations to derive task-adaptive model-merging coefficients, aiming to preserve prior knowledge while maintaining learnability of new tasks, alongside theory-driven continual meta-learning approaches such as Chen et al. (2023) that dynamically adjust update behavior under shifting environments.

**Concept drift:** Concept drift has been widely studied (Gama et al., 2014a) in the context of streaming data and evolving environments (Duda et al., 2001). Most works have focused on detecting and adapting to shifts in data distributions (Lu et al., 2018), distinguishing between sudden, incremental, and recurring drifts. Two major approaches include (i) using drift detectors (Van Looveren et al., 2024) to signal when a model should be updated to align with the new data distribution and (ii) employing online learners with forgetting mechanisms (Bifet et al., 2009) to implicitly adapt to the current state of the streaming environment. Moreover, addressing concept drift requires an understanding of its underlying causes (Krawczyk et al., 2017), whether due to changes in feature distributions (covariate shift), class boundaries (real drift), or latent dynamics in data streams. While most works in the concept drift domain have focused on shallow learning models, the deep learning community has recently started to show increasing interest (Xiang et al., 2023) in this research area.

Recent advances in concept drift and data stream mining have focused on both detection methods that scale to high-velocity streaming data and adaptation mechanisms that enable robust model updating under non-stationarity. One notable direction is unsupervised and representation-based drift detection, exemplified by the Maximum Concept Discrepancy Drift Detector (MCD-DD) (Wan et al., 2024), which leverages contrastive embeddings to detect drift without reliance on labels, outperforming classical statistical tests in high-dimensional streams. Complementary to this, Greco et al. (2025) introduced DriftLens, an unsupervised framework designed for real-time characterization of concept drift from deep feature representations, addressing limitations of existing methods in accuracy and real-time execution. Neighbor-Searching Discrepancy Drift Detection Scheme (Gu et al., 2024) isolates real concept drift by measuring classification boundary shifts between samples, a key step toward minimizing false alarms from virtual drift. On the deep learning front, DNN+AE-DD (Hu et al., 2025a) is a hybrid autoencoder and deep neural network approach that combines representation learning with reconstruction-based drift signals, demonstrating superior detection accuracy on synthetic and real-world streams compared to shallow methods. Transfer learning and multi-source strategies have also emerged, such as MARLINE (Du et al., 2025), which uses multi-source mapping to transfer knowledge in non-stationary environments and improve data stream prediction under drift. Lite-RVFL (Hu et al., 2025b) is a random vector functional-link network that adapts to concept drift without explicit detection or retraining, emphasizing recent data through exponential weighting. Harshit & Mounvik (2025) also integrated transformers with autoencoder reconstruction and multiple drift metrics to enhance early detection sensitivity and robustness.

**Concept drift in Continual Learning:** First discussion of concept drift in Continual Learning scenarios was carried out in Cossu et al. (2021) where authors suggested that excessive focus on Class-Incremental learning is too restrictive, as past classes could repeat over time in natural environments. In Korycki & Krawczyk (2021), an experience replay-based method with enhanced memory management for concept drift was introduced. It used class centroids to determine whether past samples should be relabeled. However, this approach oversimplifies concept drift by reducing the problem to two meta-labels (like vs. dislike), failing to capture the complexities of concept drift adaptation in multi-class large-scale datasets. Moreover, the authors define concept drift solely as a shift in class labels over time, overlooking the possibility of drift occurring through changes in data representations. This assumption does not align with many real-world continual learning scenarios where labels remain unchanged while data representations evolve. In Casado

et al. (2021), a novel method for federated learning was introduced that considers the possibility of concept drift, but does not explicitly measure robustness to its occurrence. In Gomez-Villa et al. (2024), the authors raise concerns about semantic drift, which causes the prototypes of learned classes to shift in feature space as new classes are introduced. Although their proposed learnable drift compensation mitigates this shift, it does not address the possibility of recurring classes or how to compensate for drift if previously seen classes reappear with altered representations.

**Concept drift vs test-time adaptation:** Recently, test-time adaptation (TTA) has attracted increasing attention from the Continual Learning community (Hong et al., 2023; Ni et al., 2025). It is important to highlight that While both concept drift and TTA address distributional shifts, they represent fundamentally distinct challenges in Continual Learning. Concept drift refers to the gradual or abrupt change in the underlying data distribution over time, necessitating continual model updates to remove outdated knowledge and update the stored past task information. In contrast, test-time adaptation focuses on rapid, often unsupervised adjustments to distribution shifts encountered during inference (Zhu et al., 2024). Crucially, concept drift emphasizes long-term updates of the past knowledge stored in the model, whereas test-time adaptation operates in a single-task setting and prioritizes immediate robustness.

**Critical gap in Continual Learning Literature:** Existing continual learning algorithms suffer from a vital limitation: they either ignore the recurrence of classes with altered representations or reduce concept drift to label-level changes, failing to capture the evolving shifts common in real-world data streams. In dynamic environments such as ecological monitoring or autonomous driving, previously seen classes can reappear under varying noise, lighting, or weather conditions, leading to representation-level drift. Our work addresses this underexplored problem by explicitly modeling concept drift changes in image representations of recurring classes.

## 3 Proposed Framework

### 3.1 Toward a Holistic Continual-Learning Paradigm

In continual learning settings, it is often assumed that once a class or task has been learned, it remains stationary over time. However, real-world environments often violate this assumption, as previously acquired knowledge may become outdated as concept drift alters the underlying distribution. Even tasks that appear stationary may shift over time because of lighting changes, seasonal effects, sensor noise, or other unforeseen factors. Consequently, a continual-learning system must not only accommodate new classes or tasks but also detect and adapt to distributional changes in classes it has already encountered. If left unaddressed, such drift renders stored representations invalid or misleading. Contemporary continual-learning methods, which focus primarily on retention, struggle in these situations and fail to adjust their predictions or internal representations to match new realities. A truly holistic and adaptive continual-learning framework must therefore revisit and revise prior knowledge whenever drift is detected, preserving relevance and accuracy for both old and new information.

### 3.2 Problem Formulation

In the context of continual learning, we formalize our problem as a sequence of tasks $\mathcal{T} = \{T_1, T_2, \ldots, T_N\}$ arriving over time. Each task $T_i$ is associated with a set of classes $\mathcal{C}_i = \{c_1^i, c_2^i, \ldots, c_{m_i}^i\}$, where $m_i$ denotes the number of classes in task $T_i$. The goal is to train a model $f_\theta : \mathcal{X} \to \mathcal{Y}$, parameterized by $\theta$, where the data distribution $\mathcal{D}$ evolves over time. The model is trained incrementally on task-specific datasets $\mathcal{D}_i = \{(x, y) \mid x \in \mathcal{X}, y \in \mathcal{Y}_i\}$, where $\mathcal{Y}_i \subseteq \mathcal{Y}$ corresponds to the labels associated with $\mathcal{C}_i$. After observing task $T_i$, the model is expected to perform well on all previously seen tasks $\{T_1, T_2, \ldots, T_i\}$.

Unlike standard class-incremental learning (CIL), where previously seen class distributions are assumed static, our setting accounts for evolving class semantics due to non-stationary environments. In other words, we extend the standard CIL setting where previously encountered classes may reappear with shifted distributions, a phenomenon known as concept drift (Gama et al., 2014b; Widmer & Kubat, 1996). Specifically, for any class $c \in \mathcal{C}_j$ from a previous task $T_j$ $(j < i)$, let $\mathcal{D}_j(c)$ denote the distribution of class $c$ in task $T_j$,

and $\mathcal{D}_i(c)$ the corresponding distribution in task $T_i$. Concept drift occurs when the class reappears in a later task with a new domain representation (Shin et al., 2017), indicating a distribution shift:

$$\mathcal{D}_i(c) \neq \mathcal{D}_j(c), \quad \text{for some } c \in \mathcal{C}_j \cap \mathcal{C}_i,$$

Given the demonstrated effectiveness of rehearsal-based continual learning methods, our proposed framework leverages memory-based strategies to address concept drift. Rehearsal methods maintain a bounded episodic memory buffer $\mathcal{M}$ of fixed capacity $|\mathcal{M}|$. For each class $c$, let $\mathcal{M}_c \subset \mathcal{M}$ denote the subset of memory allocated to class $c$. The training objective at task $T_i$ is formulated as an empirical risk over the current task data and the memory buffer:

$$\mathcal{L}(\theta) = \mathbf{E}_{(x,y) \sim \mathcal{D}_{\text{current}}}[\ell(f_\theta(x), y)] + \mathbf{E}_{(x,y) \sim \mathcal{M}}[\ell(f_\theta(x), y)],$$

where $\ell$ denotes the loss function (cross-entropy), $\mathcal{D}_{\text{current}} = \mathcal{D}_i$ is the data for the current task.

### 3.3 Concept Drift Detection

Our framework assumes a dynamic test-then-train paradigm, similar to that proposed in Bifet et al. (2010) and Sun et al. (2020), which ensures that adaptation occurs reactively in response to observable changes in the environment. Specifically, we monitor the test-time distribution of previously seen classes as they reappear. This distribution is compared against a reference distribution, which captures the historical statistics of class $c$. Only if a distributional shift is detected at test time do we proceed to adapt the model to the updated distribution $\mathcal{D}_i(c)$. This setting mirrors a data stream scenario with a dynamic and monitored test stream (Agrahari & Singh, 2022), enabling early detection and timely response to distributional shifts.

We incorporate an uncertainty-based drift detection mechanism using the two-sample Kolmogorov–Smirnov (KS) (Massey Jr, 1951) test. Let $f_\theta : \mathcal{X} \to \mathbf{R}^K$ denote a fixed pre-trained backbone that outputs class logits for input $x \in \mathcal{X}$, where $K = |\mathcal{Y}|$ is the total number of classes. The uncertainty associated with each input is quantified using predictive entropy computed from the softmax of the logits. Specifically, we define the uncertainty function as:

$$\mathcal{U}(x) = \mathbf{H}(\text{softmax}(f_\theta(x))) = -\sum_{k=1}^{K} p_k(x) \log p_k(x),$$

where $p_k(x)$ is the softmax probability for class $k$. To detect concept drift for recurring classes, we compare uncertainty distributions from two sources:

- The reference distribution $\mathcal{U}_{\text{ref}}$, computed using uncertainty values from class-specific samples stored in the memory buffer $\mathcal{M}_c$ from previous tasks.

- The test distribution $\mathcal{U}_{\text{test}}$, computed from uncertainty values of incoming samples for the same class $c$ in the current test task.

We compute the Kolmogorov–Smirnov statistic:

$$D_{\text{KS}} = \sup_u |F_{\mathcal{U}_{\text{ref}}}(u) - F_{\mathcal{U}_{\text{test}}}(u)|,$$

where $F_{\mathcal{U}}(u)$ denotes the empirical cumulative distribution function (ECDF) of uncertainty values. A concept drift event is flagged when:

$$D_{\text{KS}} > \delta,$$

where $\delta$ is a fixed threshold that governs the sensitivity of the drift detector.

### 3.4 Adaptive Memory Realignment (AMR) for Drift Adaptation

Once concept drift is detected for a class $c$, our adaptation mechanism, *Adaptive Memory Realignment (AMR)*, updates the memory buffer to reflect the new distribution. Let the memory buffer be denoted as $\mathcal{M} = \{(x_j, y_j)\}_{j=1}^{|\mathcal{M}|}$, and define the index set for class $c$ as:

$$\mathcal{I}_c = \{j \in \{1, \dots, |\mathcal{M}|\} \mid y_j = c\}$$

The adaptation process consists of the following steps:

- **Detect Drift:** Based on the KS statistic, identify the set of drifted classes $\mathcal{C}_{\text{drift}} \subseteq \mathcal{C}_i$ for the current task $T_i$.

- **Flush:** For each class $c \in \mathcal{C}_{\text{drift}}$, remove outdated samples of class $c$ from the buffer by setting $\mathcal{M}[j] = \emptyset$ for all $j \in \mathcal{I}_c$.

- **Resample:** Repopulate the freed memory slots with updated instances drawn from the new distribution $\mathcal{D}_i(c)$. For each $j \in \mathcal{I}_c$, sample a new instance $x_j^{\text{new}} \sim \mathcal{D}_i(c)$, and set $\mathcal{M}[j] = (x_j^{\text{new}}, c)$.

---

**Algorithm 1** Concept-Drift Adaptive Memory Realignment

---

1: **Input:** Task stream $\mathcal{T} = \{T_1, \ldots, T_N\}$, model $f_\theta$, memory buffer $\mathcal{M}$, drift significance threshold $\delta$
2: **Output:** Updated model $f_\theta$
3: **Initialize:** $\theta \leftarrow$ random init, $\mathcal{M} \leftarrow \emptyset$, $\mathcal{Y}_{\text{past}} \leftarrow \emptyset$
4: **for** $T_i \in \mathcal{T}$ **do**
5:     Receive data $\mathcal{D}_i = \{(x, y) \mid y \in \mathcal{Y}_i\}$
6:     **for** $c \in \mathcal{Y}_i \cap \mathcal{Y}_{\text{past}}$ **do**
7:         $\mathcal{U}_{\text{ref}} \leftarrow \{\mathcal{U}(x) \mid (x, y) \in \mathcal{M}, y = c\}$
8:         $\mathcal{U}_{\text{test}} \leftarrow \{\mathcal{U}(x) \mid (x, y) \in \mathcal{D}_i, y = c\}$
9:         $D_{\text{KS}} \leftarrow \sup_u |F_{\mathcal{U}_{\text{ref}}}(u) - F_{\mathcal{U}_{\text{test}}}(u)|$
10:        **if** $D_{\text{KS}} > \delta$ **then**                         ▷ Drift detected
11:           $\mathcal{I}_c \leftarrow \{j \in \{1, \ldots, |\mathcal{M}|\} \mid y_j = c\}$
12:           **for** $j \in \mathcal{I}_c$ **do**
13:               Sample $x_j^{\text{new}} \sim \mathcal{D}_i(c)$
14:               $\mathcal{M}[j] \leftarrow (x_j^{\text{new}}, c)$
15:           **end for**
16:        **end if**
17:     **end for**
18:     Train $f_\theta$ on $\mathcal{D}_i \cup \mathcal{M}$ with loss:

$$\mathcal{L}(\theta) = \mathbf{E}_{\mathcal{D}_i}[\ell(f_\theta(x), y)] + \mathbf{E}_{\mathcal{M}}[\ell(f_\theta(x), y)]$$

19:     $\mathcal{M} \leftarrow ReservoirSampling(\mathcal{M}, \mathcal{D}_i)$
20:     $\mathcal{Y}_{\text{past}} \leftarrow \mathcal{Y}_{\text{past}} \cup \mathcal{Y}_i$
21: **end for**
22: **return** $\theta$

---

This targeted realignment ensures that the memory buffer reflects the most recent distribution $\mathcal{D}_i(c)$ for each drifted class $c \in \mathcal{C}_{\text{drift}}$, mitigating negative transfer from outdated samples and promoting alignment with the most recent evolving distribution. Figure 2 illustrates the working principles of the proposed algorithm.

### 3.5 Theoretical Analysis

### 3.5.1 Gradient Misalignment Analysis

We now analyze why maintaining outdated representations in the memory buffer impedes learning. Consider the gradient of the loss function $\mathcal{L}(\theta)$ during rehearsal training:

$$\nabla_\theta \mathcal{L}(\theta) = \underbrace{\mathbf{E}_{(x,y) \sim \mathcal{D}_{\text{current}}}[\nabla_\theta \ell(f_\theta(x), y)]}_{\text{Current task gradient}} + \underbrace{\mathbf{E}_{(x,y) \sim \mathcal{M}}[\nabla_\theta \ell(f_\theta(x), y)]}_{\text{Rehearsal gradient}}$$

When concept drift occurs, the memory buffer contains samples from the old distribution $\mathcal{D}_j(c)$ for a drifted class $c$, while current data follows $\mathcal{D}_i(c)$.

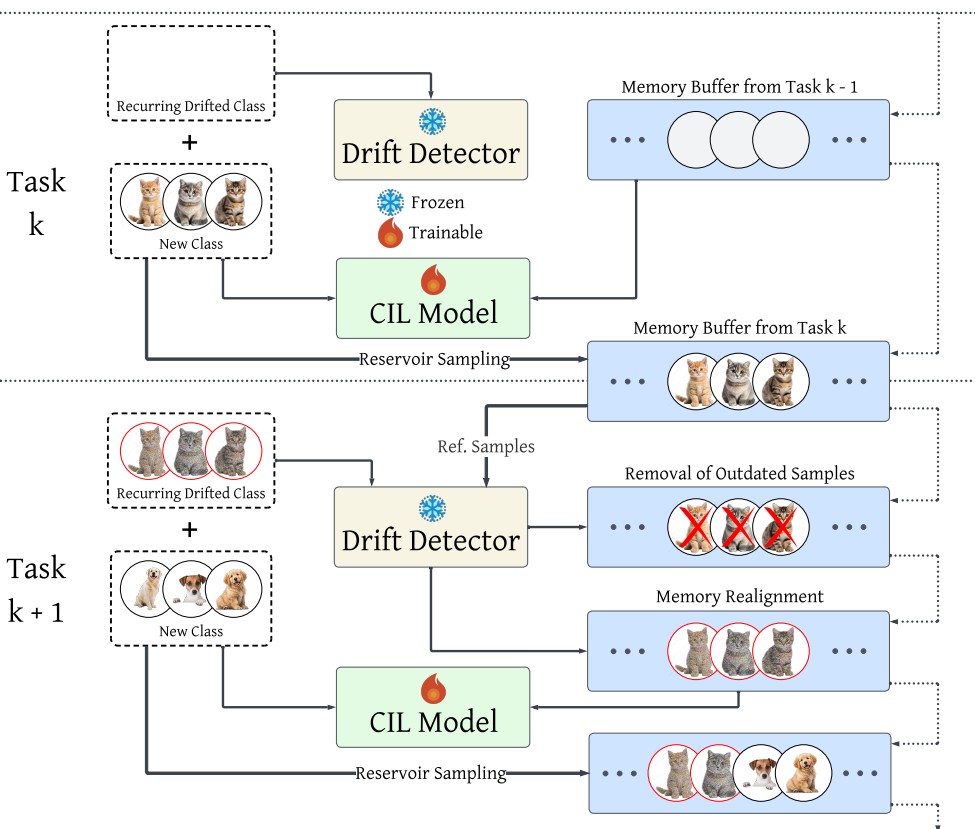

Figure 2: Flow of our proposed Concept-Drift Adaptive Memory Realignment method for continual learning under concept drift. The approach integrates an uncertainty-based drift detection module with adaptive memory management to selectively retain and update buffer samples in the presence of recurring classes exhibiting distributional shifts.

**Theorem 1:** For a drifted class $c$ with sufficiently different distributions $\mathcal{D}_j(c)$ and $\mathcal{D}_i(c)$, the expected gradients from these distributions exhibit interference, leading to misaligned parameter updates.

**Proof:** Let $G_{\text{old}} = \mathbf{E}_{(x,y) \sim \mathcal{D}_j(c)}[\nabla_\theta \ell(f_\theta(x), y)]$ and $G_{\text{new}} = \mathbf{E}_{(x,y) \sim \mathcal{D}_i(c)}[\nabla_\theta \ell(f_\theta(x), y)]$ denote the expected gradients from the old and new distributions, respectively. The cosine similarity between these gradients quantifies their alignment:

$$\text{sim}(G_{\text{old}}, G_{\text{new}}) = \frac{G_{\text{old}} \cdot G_{\text{new}}}{\|G_{\text{old}}\| \cdot \|G_{\text{new}}\|}$$

Under significant drift, this similarity decreases and can become negative. The effective gradient during training becomes:

$$G_{\text{effective}} = G_{\text{new}} + (1 - \alpha) \cdot G_{\text{old}} + \alpha \cdot G_{\text{new}}$$

where $\alpha$ represents the fraction of updated samples of class $c$ in the memory buffer. When $\alpha = 0$ (no buffer update), $G_{\text{effective}} = G_{\text{new}} + G_{\text{old}}$, which can deviate significantly from the optimal direction $G_{\text{new}}$ when $\text{sim}(G_{\text{old}}, G_{\text{new}})$ is low. To quantify this deviation, we define the gradient alignment efficiency:

$$\eta_{\text{align}} = \frac{G_{\text{effective}} \cdot G_{\text{new}}}{||G_{\text{effective}}|| \cdot ||G_{\text{new}}||}$$

For non-drifted classes, $\eta_{\text{align}} \approx 1$, indicating efficient learning. For drifted classes with outdated representations in the buffer, $\eta_{\text{align}} < 1$ and potentially $\eta_{\text{align}} \ll 1$ under severe sudden drift, as it occurs in our problem setting, resulting in inefficient learning. $\square$

To tackle this gradient misalignment, we need a better sampling strategy than reservoir sampling. This is because the probability of reservoir sampling effectively replacing the outdated samples from the memory buffer is negligibly small, as we will see in the next section.

### 3.5.2 Limitations of Conventional Reservoir Sampling

We now demonstrate why conventional reservoir sampling is suboptimal for handling concept drift compared to our targeted replacement strategy.

***Theorem 2:*** With standard reservoir sampling, the probability of effectively replacing all outdated samples of drifted classes reaches zero as the number of classes increases.

***Proof:*** For a memory buffer of size $|\mathcal{M}|$ containing $K$ classes with approximately equal representation, each class occupies approximately $|\mathcal{M}_c| \approx \frac{|\mathcal{M}|}{K}$ memory slots. For a drifted class $c$ with $n_c$ new samples, the probability that a specific old sample in the buffer is replaced under reservoir sampling is:

$$P(\text{replaced}) = 1 - \prod_{j=1}^{n_c} \left(1 - \frac{1}{|\mathcal{M}|}\right) \approx 1 - \exp\left(-\frac{n_c}{|\mathcal{M}|}\right)$$

For $n_c \ll |\mathcal{M}|$, which is typical in continual learning, this approximates to:

$$P(\text{replaced}) \approx \frac{n_c}{|\mathcal{M}|}$$

The expected number of replaced samples from class $c$ is:

$$\mathbf{E}[\text{replaced samples from } c] = |\mathcal{M}_c| \cdot P(\text{replaced}) \approx \frac{|\mathcal{M}|}{K} \cdot \frac{n_c}{|\mathcal{M}|} = \frac{n_c}{K}$$

This implies that with $n_c$ new samples and $K$ classes, standard reservoir sampling only replaces approximately $\frac{n_c}{K}$ outdated samples—far fewer than the $\frac{|\mathcal{M}|}{K}$ samples typically allocated to each class. The probability of replacing all outdated samples of class $c$ is:

$$P(\text{replace all}) = \frac{\binom{|\mathcal{M}|-|\mathcal{M}_c|}{n_c-|\mathcal{M}_c|}}{\binom{|\mathcal{M}|}{n_c}} \cdot \mathbf{1}_{n_c \geq |\mathcal{M}_c|}$$

For practical values of $|\mathcal{M}|$, $K$, and $n_c$, this probability becomes vanishingly small as the decay is combinatorial. $\square$

### 3.5.3 Optimality of Targeted Memory Realignment

In this section, we discuss why targeted buffer resampling provides the optimal gradient alignment when sudden drift occurs.

***Theorem 3:*** Targeted replacement of memory samples for drifted classes ($\alpha = 1$) maximizes gradient alignment efficiency, achieving performance comparable to training on the entire sample size of the drifted distribution.

***Proof:*** With complete targeted replacement, the effective gradient becomes:

$$G_{\text{effective}} = G_{\text{new}} + 0 \cdot G_{\text{old}} + 1 \cdot G_{\text{new}} = 2 \cdot G_{\text{new}}$$

This preserves the optimal gradient update direction while only scaling the magnitude, resulting in $\eta_{\text{align}} = 1$. Thus, our adaptation strategy ensures that gradient updates follow the same trajectory as they would if training from scratch on the new distribution. $\square$

This also ensures retention of knowledge of non-drifted classes through the memory buffer as the non-drifted concepts remain intact in the memory without the risk of potentially being replaced by random sampling.

### 3.5.4 Gradient Alignment with AMR

This section provides further justification on how the gradient alignment efficiency increases with AMR.

***Theorem 4:*** The gradient alignment efficiency $\eta_{\text{align}}$ monotonically increases with the proportion $\alpha$ of updated samples in the memory buffer, with optimal alignment achieved at $\alpha = 1$.

***Proof:*** Recall that:

$$G_{\text{effective}} = G_{\text{new}} + (1 - \alpha) \cdot G_{\text{old}} + \alpha \cdot G_{\text{new}} = (1 + \alpha) \cdot G_{\text{new}} + (1 - \alpha) \cdot G_{\text{old}}$$

The alignment efficiency is:

$$\eta_{\text{align}} = \frac{G_{\text{effective}} \cdot G_{\text{new}}}{||G_{\text{effective}}|| \cdot ||G_{\text{new}}||}$$

Substituting and simplifying:

$$\eta_{\text{align}} = \frac{(1 + \alpha)||G_{\text{new}}||^2 + (1 - \alpha)G_{\text{old}} \cdot G_{\text{new}}}{||G_{\text{effective}}|| \cdot ||G_{\text{new}}||}$$

Taking the derivative with respect to $\alpha$:

$$\frac{d\eta_{\text{align}}}{d\alpha} = \frac{||G_{\text{new}}||^2 - G_{\text{old}} \cdot G_{\text{new}}}{||G_{\text{effective}}|| \cdot ||G_{\text{new}}||} \cdot \frac{d}{d\alpha} \left( \frac{||G_{\text{effective}}||}{||G_{\text{new}}||} \right)^{-1}$$

Under the condition that $\text{sim}(G_{\text{old}}, G_{\text{new}}) < 1$, which holds under significant sudden drift, we have:

$$||G_{\text{new}}||^2 - G_{\text{old}} \cdot G_{\text{new}} > 0$$

and

$$\frac{d}{d\alpha} \left( \frac{||G_{\text{effective}}||}{||G_{\text{new}}||} \right)^{-1} > 0$$

Therefore, $\frac{d\eta_{\text{align}}}{d\alpha} > 0$, proving that the alignment efficiency increases monotonically with $\alpha$, reaching its maximum at $\alpha = 1$ (complete replacement). $\square$

### 3.5.5 Conclusion

Our mathematical analysis demonstrates that the proposed memory adaptation strategy effectively addresses concept drift in class-incremental learning by:

- Eliminating misaligned gradient interference from outdated representations

- Overcoming the limitations of conventional reservoir sampling

- Maximizing gradient alignment efficiency through targeted buffer updates

We validate our claims through targeted experiments in Section 5.1.

## 4 Experimental Setup

**Datasets:** We use standard benchmarks from the continual learning literature and adapt them to incorporate concept drift:

- **Split Fashion-MNIST** (Xiao et al., 2017): Comprises 70,000 grayscale images of size $28{\times}28$ (60,000 for training and 10,000 for testing) across 10 classes. The dataset is split into 5 tasks, each containing 2 classes.

- **Split CIFAR10 and Split CIFAR100** (Krizhevsky, 2012): Both datasets consist of 50,000 training and 10,000 test images of size 32×32. CIFAR10 is divided into 5 tasks with 2 classes per task, while CIFAR100 is divided into 10 tasks with 10 classes per task.

- **Split Tiny-ImageNet** (Le & Yang, 2015): A subset of ImageNet (Russakovsky et al., 2015) containing 100,000 training and 10,000 test images of size 64×64 across 200 classes. It is split into 10 tasks, each with 20 classes.

To induce concept drift, we apply various image transformations (Hendrycks & Dietterich, 2019) at several severity levels. As detailed in Appendix A, the highest-severity permutation provides the most pronounced distribution shift and is therefore used in all experiments.

In standard class-incremental (CIL) benchmarks, tasks contain disjoint class sets. Our framework reintroduces previously seen classes together with new ones when drift occurs. We denote these drift-augmented variants with the suffix "-CD" (for Concept Drift). To ensure a broad evaluation, we test both single and multi-drift scenarios over short and long task streams:

- **Short streams (5 tasks):** S-FMNIST-CD and S-CIFAR10-CD, each with 1 and 2 drift occurrences.

- **Long streams (10 tasks):** S-CIFAR100-CD and S-Tiny-ImageNet-CD, each with 2 and 4 drift occurrences.

Additionally, we evaluate AMR on the CLEAR-10 dataset (Lin et al., 2021) to assess performance under natural temporal drift. These real-world drift experiments are detailed in Appendix B.

**Experience replay methods:** We base our experimental evaluation around the popular rehearsal methods:

- Experience Replay (ER) (Chaudhry et al., 2019): Vanilla experience replay that revisits a subset of past samples to consolidate past knowledge while learning from new data,

- Experience Replay with Asymmetric Cross Entropy (ER-ACE) (Caccia et al., 2022): Experience replay with asymmetrical loss to reduce representation overlap of new and old classes,

- Dark Experience Replay++ (DER++) (Buzzega et al., 2020a): Experience replay with knowledge distillation from past tasks,

- Strong Experience Replay (SER) (Zhuo et al., 2023): Experience replay utilizing prediction consistency between new model mimicking future experience on current training data and old model distilling past knowledge from the memory buffer,

- Complementary Learning System-based Experience Replay (CLS-ER) (Arani et al., 2022): Experience replay with dual memories: short-term and long-term that acquire new knowledge by aligning decision boundaries with semantic memories.

**Hyperparameter and Implementation Details:** Our incremental learning framework with concept drift was implemented on top of the Mammoth library (Buzzega et al., 2020a). All experiments use a ResNet-18 backbone trained from scratch (no pre-training) with the Stochastic Gradient Descent (SGD) optimizer. We conduct additional experiments with ResNet-152 and ViT-S backbones in Appendix C to verify that AMR achieves similar drift recovery with larger and more modern architectures. The results confirm that AMR's effectiveness is architecture-independent, justifying our use of ResNet-18 for computational efficiency throughout the main experiments. Rehearsal methods utilize reservoir sampling (Vitter, 1985) for buffer management. Algorithm and dataset-specific hyperparameters are adopted from the optimal values reported by the original papers wherever possible and are detailed in Appendix D.

We omit standard augmentations during training and rehearsal, as they modify image representations and adversely affect the drift detector's performance. The drift detector relies on original image representations as a stable reference to identify image representation changes over time. For drift detection, we employ

Van Looveren et al. (2024)'s uncertainty-based detector, which uses a pre-trained ResNet-18 model with ImageNet1k weights. Statistical tests for drift detection are conducted at a significance level of 0.05.

**Metrics:** We evaluate all the methods using the following two standard metrics used in the literature:

- *FinalAverageAccuracy(FAA)* : The Final Average Accuracy measures the model's overall performance on all seen tasks after training on the entire task stream. Let $A_{i,j}$ represent the accuracy on task $j$ after training on task $i$. For a task stream with $N$ tasks, the FAA is computed as:

$$FAA = \frac{1}{N} \sum_{j=1}^{N} A_{N,j},$$

  where $A_{N,j}$ is the accuracy on task $j$ after training on the final task $N$. Higher values of $FAA$ indicate better overall retention and performance across tasks.

- *Forgetting(F)* : Forgetting quantifies the loss in performance on a task due to learning subsequent tasks. For a task $j$, the forgetting score is the difference between the accuracy immediately after learning task $j$ and its accuracy after the entire task stream:

$$F_j = \max_{k \geq j} A_{k,j} - A_{N,j},$$

  where $\max_{k \geq j} A_{k,j}$ is the highest accuracy on task $j$ during training and $A_{N,j}$ is the final accuracy on task $j$. The overall forgetting metric is the average forgetting across all tasks:

$$F = \frac{1}{N-1} \sum_{j=1}^{N-1} F_j.$$

  Lower forgetting scores indicate better retention of previously learned tasks.

## 5 Experiments

### 5.1 Empirical Validation of Theoretical Claims

To validate the theoretical claims made in section 3.5, we conduct empirical experiments comparing three adaptation strategies under concept drift:

- *Vanilla*: Baseline for a particular rehearsal method without any drift adaptation mechanism.

- *AMR (Adaptive Memory Realignment)*: Our proposed approach that selectively replaces outdated samples in the memory buffer with new instances of drifted classes, without requiring additional data for retraining.

- *Full Relearning (FR)*: A drift response that retrains the model using a full set of labeled samples from the drifted class distribution.

For each strategy, we evaluate:

- Number of labeled samples required for drift adaptation,

- Computational resource consumption in relative runtime and GFLOPs,

- Final average class-incremental accuracy after adaptation to concept drift.

These experiments are conducted on CIFAR10-CD and CIFAR100-CD datasets under different memory buffer capacities ($|\mathcal{M}| = 500$ and $5000$).

We first verify Theorem 1, which predicts gradient misalignment when a previously learned class undergoes concept drift. Our hypothesis states that if the distribution of a learned class shifts in a later task, the gradients computed from its new features will diverge from those based on the old features stored in the replay buffer. To illustrate this shift, we visualize the feature distributions of the two classes from task 1 of CIFAR10 as training progresses through five tasks. Because past training data are unavailable during class-incremental learning, we plot the test samples of task 1 after each subsequent task is learned. We use Uniform Manifold Approximation and Projection (UMAP) (McInnes & Healy, 2018) to project the high-dimensional features onto a 2D space for visualization. UMAP is a non-linear dimensionality reduction technique that preserves the local and global structure of the feature manifold, making it well-suited for tracking evolving feature distributions across tasks. Figure 3a shows that in the absence of drift, the two classes remain linearly separable. In contrast, Figure 3b depicts the scenario in which concept drift occurs at task 3. In this case, the model can no longer produce linearly separable features for the two classes using its stale buffer, leading to overlapping representations. Without adaptation, such overlap yields sub-optimal gradient updates and a drop in accuracy on the drifted classes, which supports our claim in Theorem 1.

Figure 4 highlights the computational efficiency trends of AMR in terms of both relative sample requirement (4a), runtime (4b), and FLOPs (4c). While FR requires forward-backward passes on a large number of labeled examples, AMR limits adaptation to small, targeted memory realignments. This validates Theorem 3, which showed that full replacement of drifted samples restores optimal gradient alignment without the cost of full-scale retraining.

As shown in Figure 5, AMR closely matches the accuracy improvements of FR for both single and multiple drift occurrences but achieves this with a substantially reduced sample requirement. This supports Theorem 4, which predicts increasing gradient alignment with targeted memory updates, leading to efficient drift recovery. As a result, AMR enables efficient adaptation to concept drift without the need for extensive retraining, offering a resource-efficient solution for real-world continual learning scenarios where drift is prevalent.

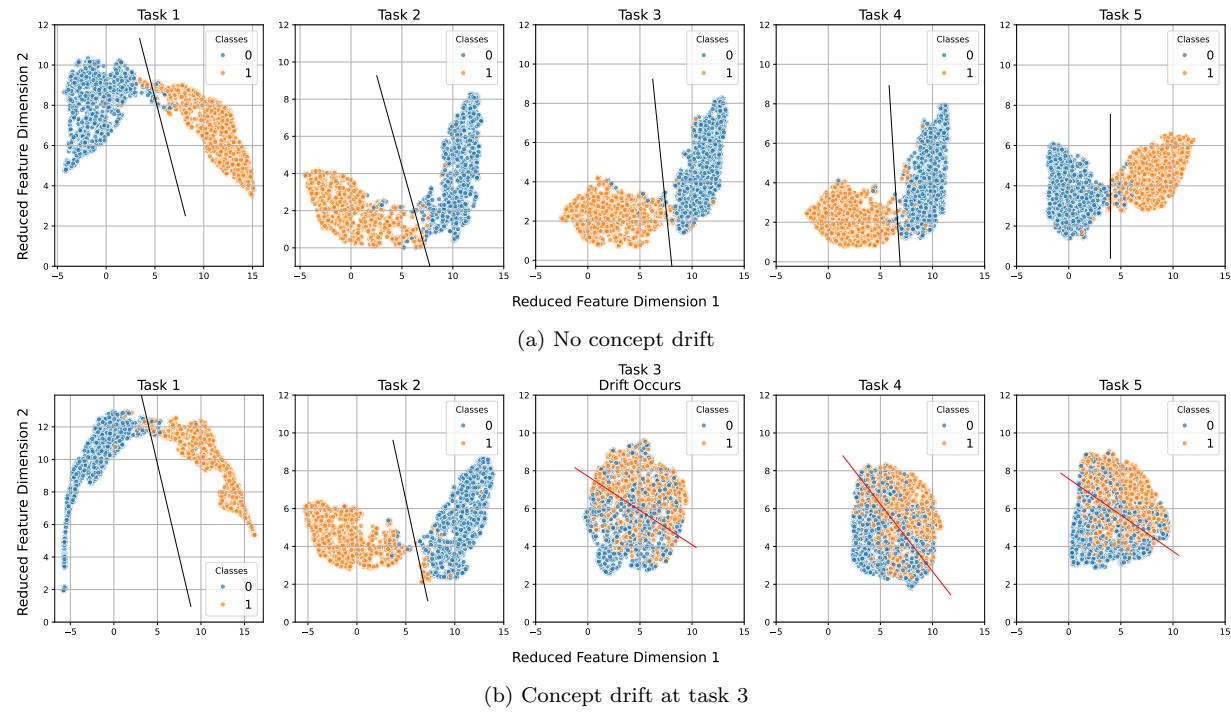

(a) No concept drift

(b) Concept drift at task 3

Figure 3: Evolution of the task-1 feature space (two classes) across five tasks on CIFAR-10. Without drift (top) the classes remain linearly separable; with drift introduced at task 3 (bottom) the features collapse.

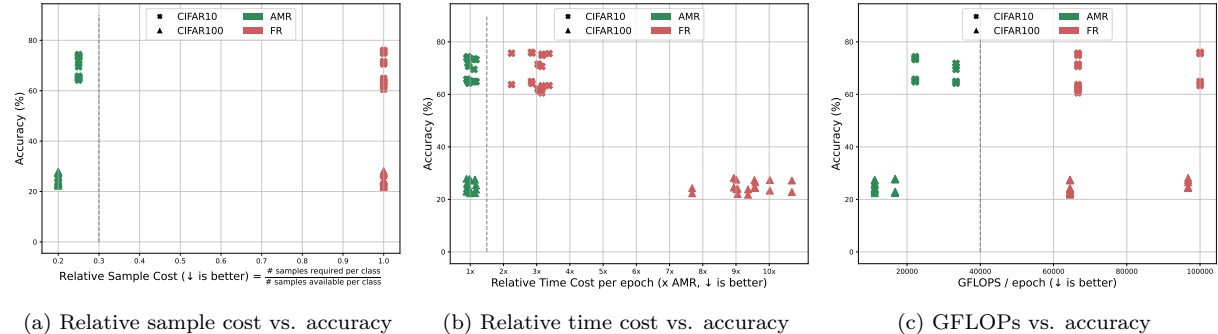

(a) Relative sample cost vs. accuracy     (b) Relative time cost vs. accuracy     (c) GFLOPs vs. accuracy

Figure 4: Comparison of computational cost and accuracy for different drift adaptation strategies. AMR achieves near-equivalent accuracy to FR with significantly lower sample requirement, relative time and GFLOP consumption.

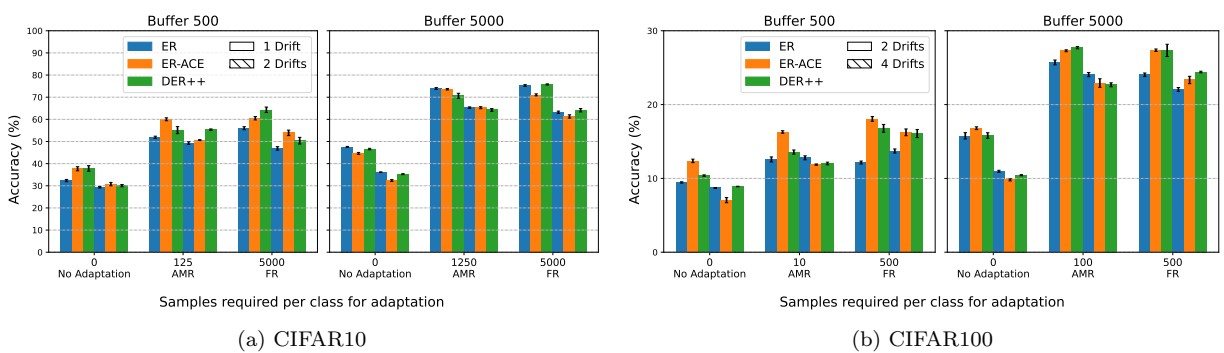

(a) CIFAR10                     (b) CIFAR100

Figure 5: Comparison of class-incremental accuracy across different experience replay methods and varying number of drifts using *No Adaptation*, *AMR*, and *Full Relearning* strategies. AMR consistently achieves comparable accuracy to FR while using significantly fewer labeled samples.

## 5.2 Experimental Results

We conducted a series of experiments on Fashion-MNIST-CD, CIFAR10-CD, CIFAR100-CD, and Tiny-ImageNet-CD to evaluate the effectiveness of our proposed method under concept drift. The experiments varied buffer sizes ($|\mathcal{M}| = 500$ and $5000$) and included both single and multiple drift scenarios. An overview of the experimental outcomes is provided in Tables 1, 2, 3 and 4.

Figures 6, 7, 8, and 9 present the results on shorter task streams from Tables 1 and 2 on Fashion-MNIST-CD and CIFAR10-CD under one and two drift events. Across all settings, both the AMR and FR strategies consistently restore model performance following drift(s). These results confirm that the proposed AMR strategy can match the performance of FR. Interestingly, we observe that larger buffers exacerbate the impact of concept drift. We hypothesize that smaller buffers, due to stronger forgetting, reduce reliance on outdated representations, forcing the model to adapt more aggressively to new data. In contrast, larger buffers retain older samples longer, potentially hindering adaptation by reinforcing outdated features. This observation reveals an intriguing insight that while larger buffers improve performance in static continual learning settings, they may require adaptive mechanisms like AMR to remain effective under drift.

To further evaluate generalization, we test our approach on CIFAR-100-CD and Tiny-ImageNet-CD—benchmarks with longer task streams and larger class spaces (Tables 3 and 4). We simulate two and four drift events, each of which changes the representations of all previously seen classes. Figures 10, 11, and 12 show that, while FR consistently restores full performance, AMR achieves comparable accuracy and reliably outperforms the No-Adaptation baseline.

On large datasets such as Tiny-ImageNet-CD, forgetting is so pronounced that a small buffer cannot support meaningful recovery. Table 3 confirms that a 500-sample buffer suffices for CIFAR-100-CD. However, from

Table 1: Final Average Accuracy (FAA[↑]) and Forgetting (F[↓]) for Split-Fashion-MNIST-CD (3-run average).

*Vanilla = Baseline without Drift Adaptation, FR = Full Relearning, AMR = Adaptive Memory Realignment*

| | | | **S-FMNIST-CD** | | |
|---|---|---|---|---|---|
| | | | No Drift | 1 Drift | 2 Drifts |
| $\mathcal{M}$ | Method | Adaptation | FAA↑$_{\pm std}$ (F↓) | FAA↑$_{\pm std}$ (F↓) | FAA↑$_{\pm std}$ (F↓) |
| 500 | ER | Vanilla | 74.35$_{\pm1.27}$ (22.86) | 63.01$_{\pm0.66}$ (36.69) | 54.61$_{\pm0.86}$ (55.03) |
| | | FR | - | 84.26$_{\pm0.49}$ (18.27) | 82.23$_{\pm0.19}$ (20.50) |
| | | AMR | - | 86.30$_{\pm0.86}$ (15.88) | 78.80$_{\pm0.34}$ (24.79) |
| | ER-ACE | Vanilla | 82.77$_{\pm0.14}$ (8.32) | 73.00$_{\pm1.67}$ (20.71) | 63.13$_{\pm1.20}$ (31.75) |
| | | FR | - | 87.83$_{\pm0.16}$ (1.40) | 89.65$_{\pm0.50}$ (5.49) |
| | | AMR | - | 88.67$_{\pm0.93}$ (6.01) | 85.11$_{\pm0.35}$ (12.80) |
| | DER++ | Vanilla | 81.92$_{\pm0.07}$ (14.07) | 69.69$_{\pm0.87}$ (27.07) | 60.13$_{\pm1.37}$ (40.40) |
| | | FR | - | 89.13$_{\pm0.42}$ (7.93) | 74.39$_{\pm0.41}$ (26.78) |
| | | AMR | - | 88.28$_{\pm0.90}$ (10.27) | 79.86$_{\pm0.15}$ (18.93) |
| | SER | Vanilla | 81.38$_{\pm0.47}$ (13.19) | 70.32$_{\pm0.99}$ (26.39) | 63.18$_{\pm1.43}$ (34.82) |
| | | FR | - | 86.86$_{\pm0.15}$ (7.38) | 89.37$_{\pm0.27}$ (3.40) |
| | | AMR | - | 89.39$_{\pm0.15}$ (6.04) | 80.25$_{\pm0.60}$ (16.68) |
| | CLS-ER | Vanilla | 79.98$_{\pm0.72}$ (18.82) | 68.13$_{\pm1.64}$ (33.45) | 57.34$_{\pm1.40}$ (46.85) |
| | | FR | - | 76.52$_{\pm0.29}$ (24.44) | 85.82$_{\pm0.18}$ (12.93) |
| | | AMR | - | 76.37$_{\pm0.61}$ (25.30) | 79.11$_{\pm0.63}$ (21.28) |
| 5000 | ER | Vanilla | 80.62$_{\pm0.97}$ (17.55) | 62.51$_{\pm1.20}$ (40.55) | 58.79$_{\pm0.32}$ (48.08) |
| | | FR | - | 85.11$_{\pm1.17}$ (10.24) | 90.76$_{\pm0.45}$ (7.89) |
| | | AMR | - | 84.51$_{\pm0.64}$ (14.19) | 92.48$_{\pm0.12}$ (6.80) |
| | ER-ACE | Vanilla | 87.14$_{\pm0.27}$ (3.58) | 74.89$_{\pm0.65}$ (18.04) | 58.51$_{\pm1.09}$ (39.78) |
| | | FR | - | 89.23$_{\pm0.73}$ (0.84) | 90.40$_{\pm0.57}$ (4.44) |
| | | AMR | - | 93.39$_{\pm0.18}$ (2.75) | 93.17$_{\pm0.11}$ (4.25) |
| | DER++ | Vanilla | 87.99$_{\pm0.13}$ (5.88) | 74.27$_{\pm1.16}$ (23.23) | 62.65$_{\pm1.16}$ (37.55) |
| | | FR | - | 90.70$_{\pm0.61}$ (4.83) | 91.23$_{\pm0.07}$ (1.44) |
| | | AMR | - | 91.82$_{\pm0.28}$ (5.73) | 87.72$_{\pm0.77}$ (10.98) |
| | SER | Vanilla | 87.50$_{\pm0.22}$ (3.78) | 72.86$_{\pm0.57}$ (22.27) | 66.19$_{\pm1.14}$ (30.83) |
| | | FR | - | 89.08$_{\pm0.43}$ (1.14) | 90.66$_{\pm0.24}$ (3.54) |
| | | AMR | - | 91.39$_{\pm0.79}$ (6.13) | 91.51$_{\pm0.23}$ (4.98) |
| | CLS-ER | Vanilla | 78.17$_{\pm1.63}$ (22.24) | 59.91$_{\pm1.13}$ (43.87) | 59.80$_{\pm1.48}$ (44.19) |
| | | FR | - | 87.37$_{\pm0.13}$ (10.93) | 85.48$_{\pm0.95}$ (11.34) |
| | | AMR | - | 87.44$_{\pm0.45}$ (11.92) | 80.49$_{\pm0.44}$ (20.93) |

Table 2: Final Average Accuracy (FAA[↑]) and Forgetting (F[↓]) for Split-CIFAR10-CD (3-run average).

*Vanilla = Baseline without Drift Adaptation, FR = Full Relearning, AMR = Adaptive Memory Realignment*

| | | | **S-CIFAR10-CD** | | |
|---|---|---|---|---|---|
| | | | No Drift | 1 Drift | 2 Drifts |
| $\mathcal{M}$ | Method | Adaptation | FAA↑$_{\pm std}$ (F↓) | FAA↑$_{\pm std}$ (F↓) | FAA↑$_{\pm std}$ (F↓) |
| 500 | ER | Vanilla | 34.37$_{\pm0.50}$ (74.96) | 32.36$_{\pm0.51}$ (77.06) | 29.32$_{\pm0.31}$ (81.53) |
| | | FR | - | 56.00$_{\pm0.69}$ (47.00) | 46.88$_{\pm0.80}$ (58.06) |
| | | AMR | - | 51.88$_{\pm0.52}$ (52.44) | 49.25$_{\pm0.60}$ (55.78) |
| | ER-ACE | Vanilla | 57.32$_{\pm1.10}$ (29.26) | 37.68$_{\pm0.93}$ (52.40) | 30.76$_{\pm0.77}$ (64.03) |
| | | FR | - | 60.50$_{\pm0.76}$ (27.72) | 53.90$_{\pm1.24}$ (37.75) |
| | | AMR | - | 60.02$_{\pm0.67}$ (31.63) | 50.62$_{\pm0.21}$ (44.63) |
| | DER++ | Vanilla | 41.40$_{\pm0.62}$ (63.55) | 37.83$_{\pm1.24}$ (68.14) | 30.07$_{\pm0.51}$ (78.15) |
| | | FR | - | 64.31$_{\pm1.24}$ (35.21) | 50.36$_{\pm1.51}$ (51.62) |
| | | AMR | - | 55.13$_{\pm1.60}$ (46.51) | 55.36$_{\pm0.31}$ (46.55) |
| | SER | Vanilla | 58.98$_{\pm1.05}$ (29.73) | 40.88$_{\pm0.86}$ (53.20) | 32.54$_{\pm0.70}$ (63.96) |
| | | FR | - | 68.15$_{\pm1.14}$ (22.05) | 59.60$_{\pm0.69}$ (23.66) |
| | | AMR | - | 62.55$_{\pm0.88}$ (32.45) | 48.68$_{\pm1.45}$ (51.50) |
| | CLS-ER | Vanilla | 28.78$_{\pm0.61}$ (81.80) | 29.20$_{\pm0.72}$ (81.05) | 26.20$_{\pm0.42}$ (84.63) |
| | | FR | - | 54.46$_{\pm0.31}$ (49.34) | 46.48$_{\pm0.75}$ (58.90) |
| | | AMR | - | 54.06$_{\pm0.71}$ (49.85) | 53.41$_{\pm0.84}$ (50.61) |
| 5000 | ER | Vanilla | 66.17$_{\pm0.54}$ (33.25) | 47.56$_{\pm0.22}$ (56.36) | 36.13$_{\pm0.15}$ (71.00) |
| | | FR | - | 75.27$_{\pm0.41}$ (21.78) | 63.20$_{\pm0.52}$ (36.73) |
| | | AMR | - | 73.92$_{\pm0.42}$ (24.33) | 65.27$_{\pm0.36}$ (35.24) |
| | ER-ACE | Vanilla | 68.76$_{\pm0.57}$ (13.86) | 44.62$_{\pm0.44}$ (46.05) | 32.37$_{\pm0.43}$ (59.81) |
| | | FR | - | 71.07$_{\pm0.45}$ (15.28) | 61.31$_{\pm0.75}$ (27.70) |
| | | AMR | - | 73.58$_{\pm0.32}$ (15.72) | 65.27$_{\pm0.46}$ (28.43) |
| | DER++ | Vanilla | 65.17$_{\pm0.95}$ (29.63) | 46.58$_{\pm0.33}$ (52.06) | 35.25$_{\pm0.23}$ (67.99) |
| | | FR | - | 75.76$_{\pm0.25}$ (19.28) | 64.12$_{\pm0.77}$ (31.77) |
| | | AMR | - | 70.65$_{\pm1.13}$ (26.54) | 64.24$_{\pm0.62}$ (35.23) |
| | SER | Vanilla | 69.22$_{\pm0.33}$ (15.20) | 46.73$_{\pm0.44}$ (43.97) | 33.95$_{\pm0.40}$ (58.70) |
| | | FR | - | 72.74$_{\pm0.76}$ (12.13) | 63.54$_{\pm0.38}$ (23.94) |
| | | AMR | - | 72.96$_{\pm0.43}$ (17.36) | 59.05$_{\pm0.68}$ (39.98) |
| | CLS-ER | Vanilla | 66.87$_{\pm0.77}$ (33.13) | 49.06$_{\pm0.17}$ (55.47) | 36.31$_{\pm0.23}$ (71.57) |
| | | FR | - | 76.27$_{\pm0.11}$ (21.16) | 64.59$_{\pm0.30}$ (34.99) |
| | | AMR | - | 77.20$_{\pm0.42}$ (20.18) | 67.58$_{\pm0.40}$ (32.15) |

Table 4, it is evident that several methods fail to recover on Tiny-ImageNet-CD with the same capacity. We therefore recommend a buffer size of 5000 for effective drift adaptation on large-scale datasets.

Table 3: Final Average Accuracy (FAA[↑]) and Forgetting (F[↓]) for Split-CIFAR100-CD (3-run average).

*Vanilla = Baseline without Drift Adaptation, FR = Full Relearning, AMR = Adaptive Memory Realignment*

| | | | S-CIFAR100-CD | | |
|---|---|---|---|---|---|
| $\mathcal{M}$ | *Method* | *Adaptation* | *No Drift* FAA↑±std (F↓) | *2 Drifts* FAA↑±std (F↓) | *4 Drifts* FAA↑±std (F↓) |
| 500 | ER | *Vanilla* | $9.74_{\pm 0.18}$ (74.80) | $9.45_{\pm 0.09}$ (74.94) | $8.70_{\pm 0.07}$ (75.58) |
| | | *FR* | - | $12.15_{\pm 0.20}$ (69.16) | $13.71_{\pm 0.28}$ (66.74) |
| | | *AMR* | - | $12.57_{\pm 0.34}$ (70.55) | $12.79_{\pm 0.27}$ (69.96) |
| | ER-ACE | *Vanilla* | $21.95_{\pm 0.35}$ (39.40) | $12.37_{\pm 0.24}$ (48.92) | $7.05_{\pm 0.35}$ (55.04) |
| | | *FR* | - | $18.06_{\pm 0.32}$ (43.28) | $16.24_{\pm 0.46}$ (49.96) |
| | | *AMR* | - | $16.29_{\pm 0.17}$ (45.09) | $11.87_{\pm 0.09}$ (51.70) |
| | DER++ | *Vanilla* | $12.11_{\pm 0.07}$ (69.80) | $10.38_{\pm 0.10}$ (72.03) | $8.90_{\pm 0.04}$ (74.33) |
| | | *FR* | - | $16.78_{\pm 0.53}$ (63.39) | $16.08_{\pm 0.55}$ (63.68) |
| | | *AMR* | - | $13.57_{\pm 0.28}$ (67.61) | $12.02_{\pm 0.18}$ (68.91) |
| | SER | *Vanilla* | $26.60_{\pm 0.28}$ (42.38) | $18.62_{\pm 0.11}$ (51.06) | $12.29_{\pm 0.15}$ (58.29) |
| | | *FR* | - | $20.71_{\pm 0.14}$ (41.83) | $22.00_{\pm 0.48}$ (42.04) |
| | | *AMR* | - | $17.73_{\pm 0.71}$ (46.52) | $13.37_{\pm 0.18}$ (52.77) |
| | CLS-ER | *Vanilla* | $11.85_{\pm 0.02}$ (73.15) | $10.45_{\pm 0.12}$ (74.87) | $9.38_{\pm 0.13}$ (76.09) |
| | | *FR* | - | $18.99_{\pm 0.14}$ (62.31) | $17.04_{\pm 0.16}$ (64.99) |
| | | *AMR* | - | $17.11_{\pm 0.10}$ (66.79) | $14.95_{\pm 0.31}$ (68.71) |
| 5000 | ER | *Vanilla* | $22.89_{\pm 0.20}$ (57.28) | $15.77_{\pm 0.43}$ (64.33) | $10.96_{\pm 0.14}$ (69.67) |
| | | *FR* | - | $24.04_{\pm 0.24}$ (53.18) | $22.05_{\pm 0.26}$ (54.66) |
| | | *AMR* | - | $25.71_{\pm 0.33}$ (53.43) | $24.06_{\pm 0.29}$ (55.40) |
| | ER-ACE | *Vanilla* | $32.44_{\pm 0.45}$ (31.83) | $16.80_{\pm 0.19}$ (48.50) | $9.80_{\pm 0.14}$ (56.52) |
| | | *FR* | - | $27.36_{\pm 0.17}$ (38.41) | $23.33_{\pm 0.50}$ (44.47) |
| | | *AMR* | - | $27.29_{\pm 0.13}$ (39.81) | $22.92_{\pm 0.58}$ (46.73) |
| | DER++ | *Vanilla* | $30.88_{\pm 0.33}$ (36.13) | $15.81_{\pm 0.38}$ (52.48) | $10.43_{\pm 0.08}$ (58.87) |
| | | *FR* | - | $27.34_{\pm 0.83}$ (40.33) | $24.41_{\pm 0.11}$ (44.11) |
| | | *AMR* | - | $27.71_{\pm 0.15}$ (45.62) | $22.68_{\pm 0.27}$ (54.97) |
| | SER | *Vanilla* | $36.18_{\pm 0.61}$ (14.16) | $17.19_{\pm 0.41}$ (34.61) | $10.96_{\pm 0.16}$ (42.29) |
| | | *FR* | - | $26.79_{\pm 0.58}$ (25.56) | $25.33_{\pm 0.38}$ (28.76) |
| | | *AMR* | - | $27.40_{\pm 0.24}$ (34.61) | $20.88_{\pm 0.51}$ (46.44) |
| | CLS-ER | *Vanilla* | $32.78_{\pm 0.14}$ (43.26) | $19.43_{\pm 0.12}$ (57.74) | $12.83_{\pm 0.19}$ (64.49) |
| | | *FR* | - | $32.06_{\pm 0.08}$ (42.17) | $26.47_{\pm 0.21}$ (48.64) |
| | | *AMR* | - | $31.47_{\pm 0.31}$ (44.51) | $26.75_{\pm 0.37}$ (51.58) |

Table 4: Final Average Accuracy (FAA[↑]) and Forgetting (F[↓]) for Split-Tiny-ImageNet-CD (3-run average).

*Vanilla = Baseline without Drift Adaptation, FR = Full Relearning, AMR = Adaptive Memory Realignment*

| | | | S-Tiny-ImageNet-CD | | |
|---|---|---|---|---|---|
| $\mathcal{M}$ | *Method* | *Adaptation* | *No Drift* FAA↑±std (F↓) | *2 Drifts* FAA↑±std (F↓) | *4 Drifts* FAA↑±std (F↓) |
| 500 | ER | *Vanilla* | $6.22_{\pm 0.11}$ (57.50) | $6.25_{\pm 0.15}$ (58.50) | $6.30_{\pm 0.03}$ (58.16) |
| | | *FR* | - | $6.64_{\pm 0.15}$ (55.21) | $6.48_{\pm 0.17}$ (55.67) |
| | | *AMR* | - | $6.19_{\pm 0.15}$ (57.67) | $6.18_{\pm 0.10}$ (57.17) |
| | ER-ACE | *Vanilla* | $10.76_{\pm 0.13}$ (32.76) | $5.00_{\pm 0.18}$ (37.99) | $2.56_{\pm 0.05}$ (41.23) |
| | | *FR* | - | $6.00_{\pm 0.37}$ (39.87) | $5.46_{\pm 0.23}$ (44.64) |
| | | *AMR* | - | $5.96_{\pm 0.20}$ (38.40) | $3.19_{\pm 0.11}$ (41.38) |
| | DER++ | *Vanilla* | $6.61_{\pm 0.10}$ (58.85) | $6.45_{\pm 0.12}$ (58.67) | $6.44_{\pm 0.07}$ (59.03) |
| | | *FR* | - | $7.04_{\pm 0.19}$ (48.77) | $7.97_{\pm 0.11}$ (54.27) |
| | | *AMR* | - | $6.65_{\pm 0.24}$ (56.54) | $6.20_{\pm 0.08}$ (57.37) |
| | SER | *Vanilla* | $16.41_{\pm 0.59}$ (21.69) | $10.74_{\pm 0.26}$ (27.80) | $6.86_{\pm 0.15}$ (32.29) |
| | | *FR* | - | $11.71_{\pm 0.12}$ (26.65) | $8.71_{\pm 0.33}$ (31.77) |
| | | *AMR* | - | $7.32_{\pm 0.29}$ (28.07) | $4.73_{\pm 0.22}$ (26.84) |
| | CLS-ER | *Vanilla* | $6.50_{\pm 0.07}$ (57.30) | $6.30_{\pm 0.00}$ (57.60) | $6.16_{\pm 0.27}$ (57.21) |
| | | *FR* | - | $7.87_{\pm 0.04}$ (53.70) | $8.72_{\pm 0.03}$ (53.56) |
| | | *AMR* | - | $7.46_{\pm 0.19}$ (56.56) | $7.30_{\pm 0.08}$ (56.29) |
| 5000 | ER | *Vanilla* | $9.91_{\pm 0.26}$ (58.40) | $8.30_{\pm 0.06}$ (60.74) | $7.12_{\pm 0.14}$ (61.27) |
| | | *FR* | - | $9.69_{\pm 0.05}$ (56.20) | $9.35_{\pm 0.17}$ (56.19) |
| | | *AMR* | - | $10.12_{\pm 0.08}$ (57.86) | $8.57_{\pm 0.18}$ (58.20) |
| | ER-ACE | *Vanilla* | $16.16_{\pm 0.30}$ (32.52) | $8.60_{\pm 0.09}$ (40.89) | $5.16_{\pm 0.17}$ (44.78) |
| | | *FR* | - | $11.71_{\pm 0.04}$ (38.26) | $9.23_{\pm 0.05}$ (45.20) |
| | | *AMR* | - | $11.10_{\pm 0.17}$ (41.24) | $7.48_{\pm 0.08}$ (46.96) |
| | DER++ | *Vanilla* | $11.55_{\pm 0.45}$ (36.43) | $6.52_{\pm 0.08}$ (42.46) | $5.38_{\pm 0.15}$ (43.10) |
| | | *FR* | - | $10.52_{\pm 0.30}$ (39.23) | $8.82_{\pm 0.12}$ (41.06) |
| | | *AMR* | - | $9.87_{\pm 0.62}$ (49.14) | $8.04_{\pm 0.17}$ (53.03) |
| | SER | *Vanilla* | $16.22_{\pm 0.45}$ (10.87) | $7.33_{\pm 0.29}$ (20.35) | $4.84_{\pm 0.29}$ (23.95) |
| | | *FR* | - | $9.46_{\pm 0.38}$ (20.87) | $7.63_{\pm 0.04}$ (21.91) |
| | | *AMR* | - | $10.53_{\pm 0.22}$ (23.22) | $6.84_{\pm 0.26}$ (31.33) |
| | CLS-ER | *Vanilla* | $16.23_{\pm 0.38}$ (39.80) | $9.81_{\pm 0.16}$ (46.77) | $7.41_{\pm 0.13}$ (49.77) |
| | | *FR* | - | $14.47_{\pm 0.18}$ (42.63) | $11.13_{\pm 0.14}$ (47.10) |
| | | *AMR* | - | $14.33_{\pm 0.29}$ (43.24) | $10.78_{\pm 0.30}$ (48.96) |

Beyond accuracy, practical deployment requires minimizing both the number of labeled samples and the computational cost of adaptation. To quantify these trade-offs, we compared the two approaches across three key

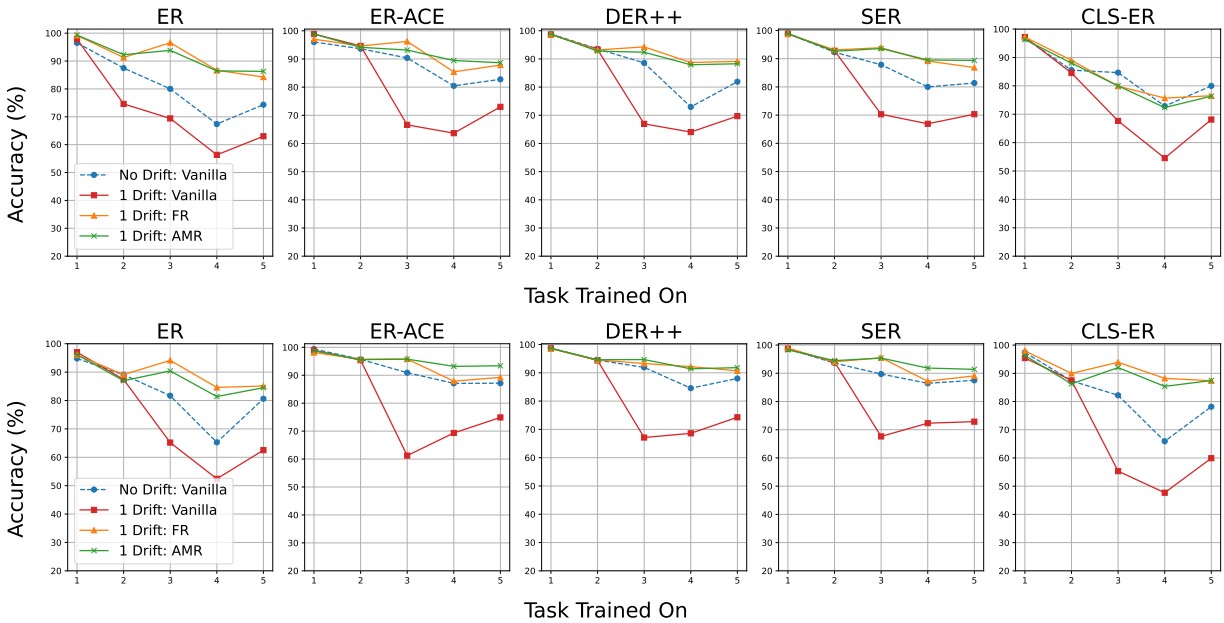

Figure 6: Class-incremental accuracy on S-FashionMNIST-CD with a single drift event occurring at task 3. Results are shown for buffer sizes 500 (top) and 5000 (bottom).

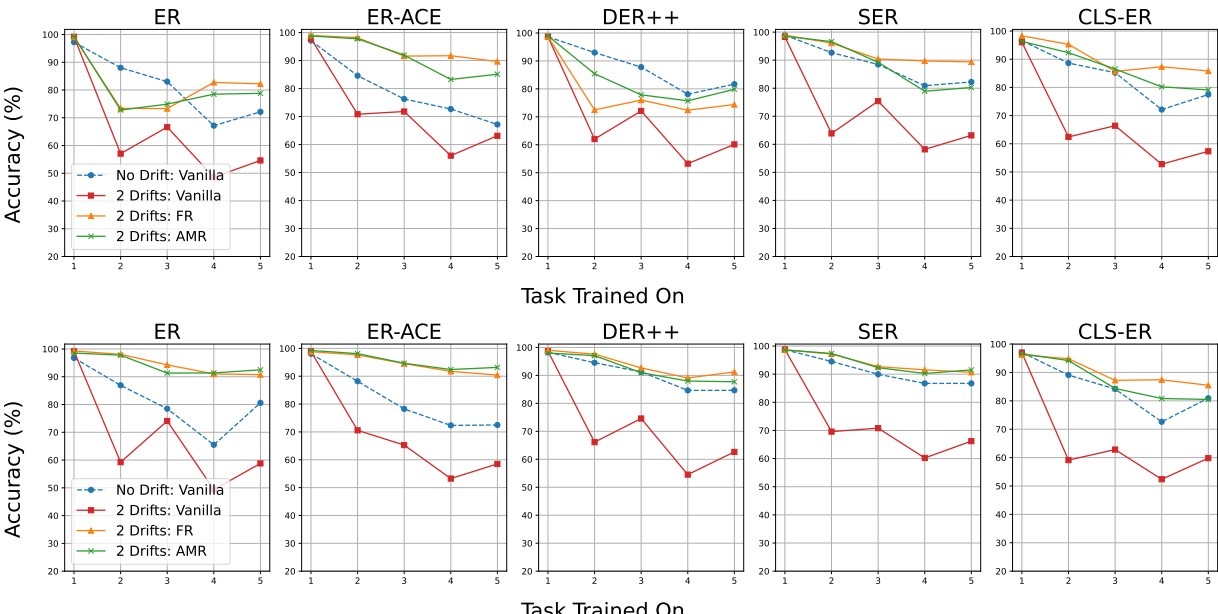

Figure 7: Class-incremental accuracy on S-FashionMNIST-CD with drift events at tasks 2 and 4. Results are shown for buffer sizes 500 (top) and 5000 (bottom).

metrics: time per epoch, GFLOPs, and the number of labeled samples required for adaptation. All metrics were normalized with respect to the FR baseline, which is assigned a normalized value of 1.0 (representing the highest cost). The performance of AMR is expressed on a 0∼1 scale, where lower values indicate better efficiency relative to FR. As shown in Figure 13, AMR consistently requires fewer computational resources and labeled samples across FashionMNIST-CD, CIFAR10-CD, CIFAR100-CD, and Tiny-ImageNet-CD datasets.

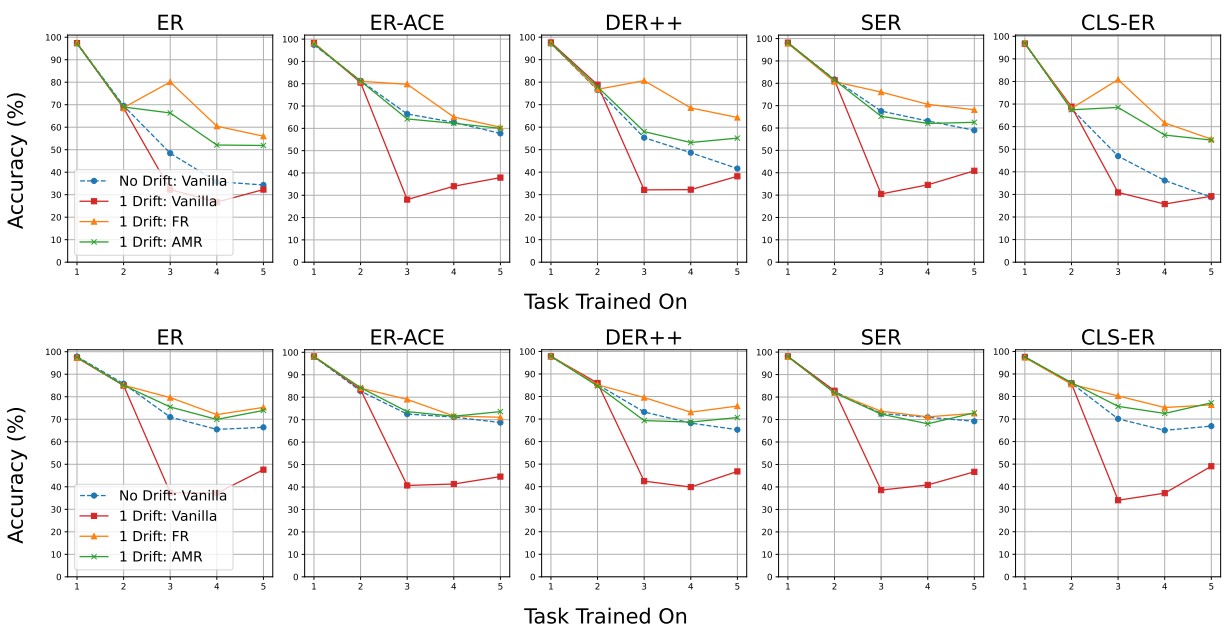

Figure 8: Class-incremental accuracy on S-CIFAR10-CD with a single drift event occurring at task 3. Results are shown for buffer sizes 500 (top) and 5000 (bottom).

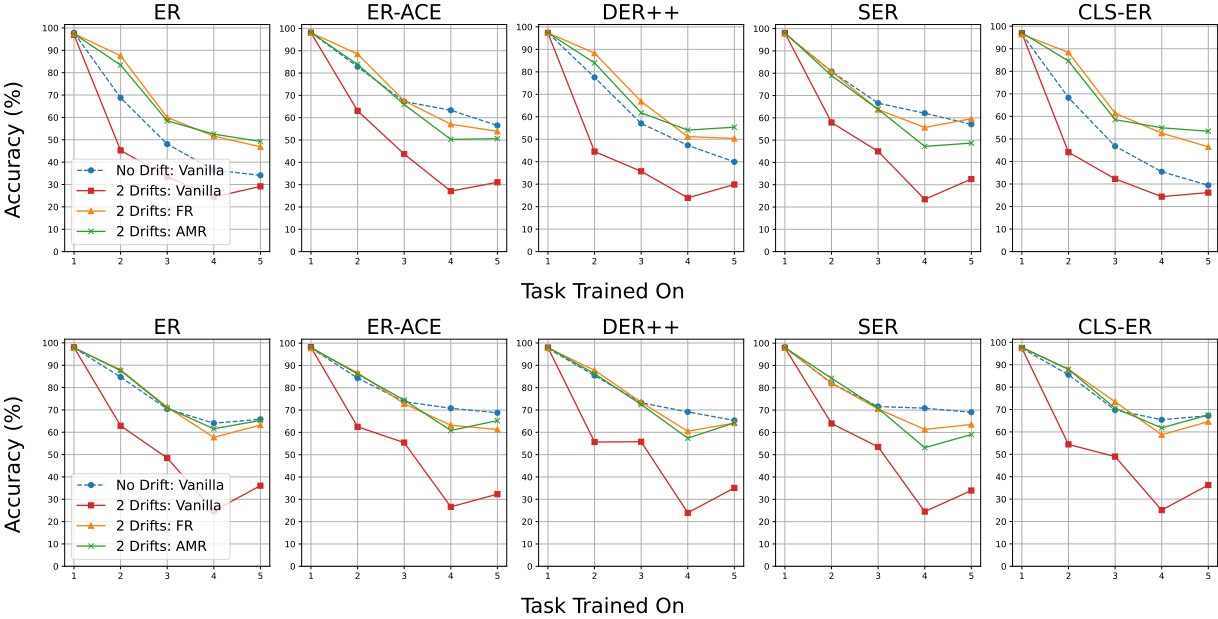

Figure 9: Class-incremental accuracy on S-CIFAR10-CD with drift events at tasks 2 and 4. Results are shown for buffer sizes 500 (top) and 5000 (bottom).

As expected, these results confirm that although effective at mitigating concept drift, the FR strategy incurs significant overhead, as it requires full retraining on large labeled datasets following each drift event. In contrast, AMR's selective replacement of only outdated entries in the memory buffer avoids unnecessary retraining and reduces the overall adaptation cost. This makes AMR not only competitive in terms of accuracy, but also significantly more efficient in both computational and labeling costs. By operating well below the resource demands of Full Relearning, AMR presents a practical and scalable solution for real-world continual learning under concept drift.

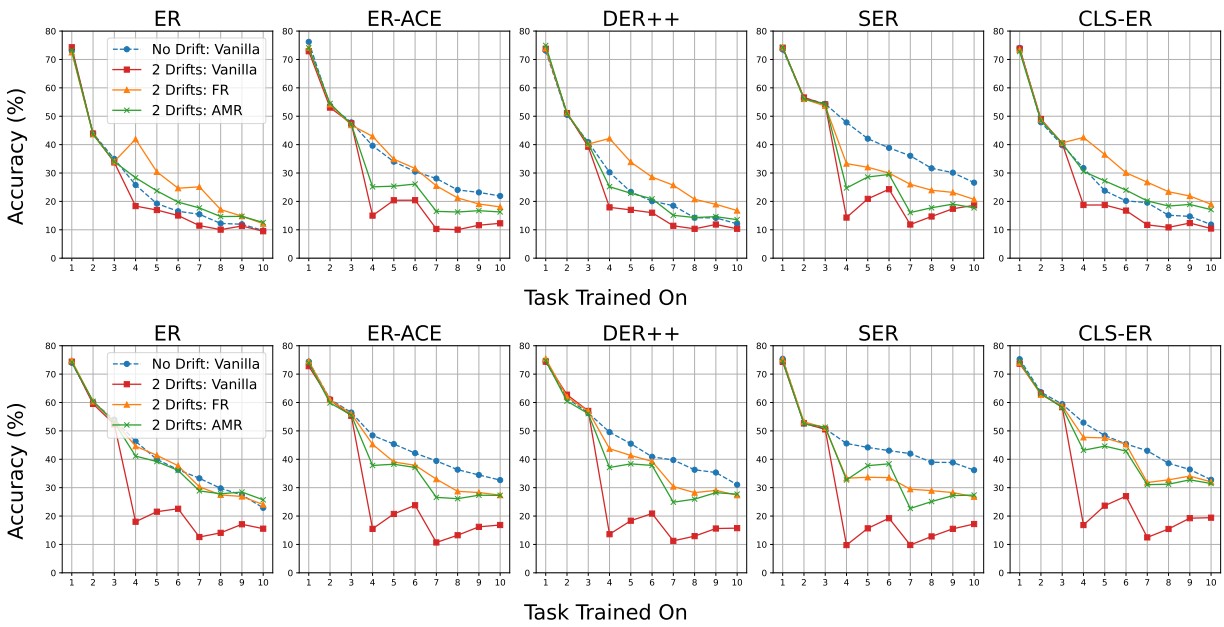

Figure 10: Class-incremental accuracy on S-CIFAR100 with drift events at tasks 4 and 7. Results are shown for buffer sizes 500 (top) and 5000 (bottom).

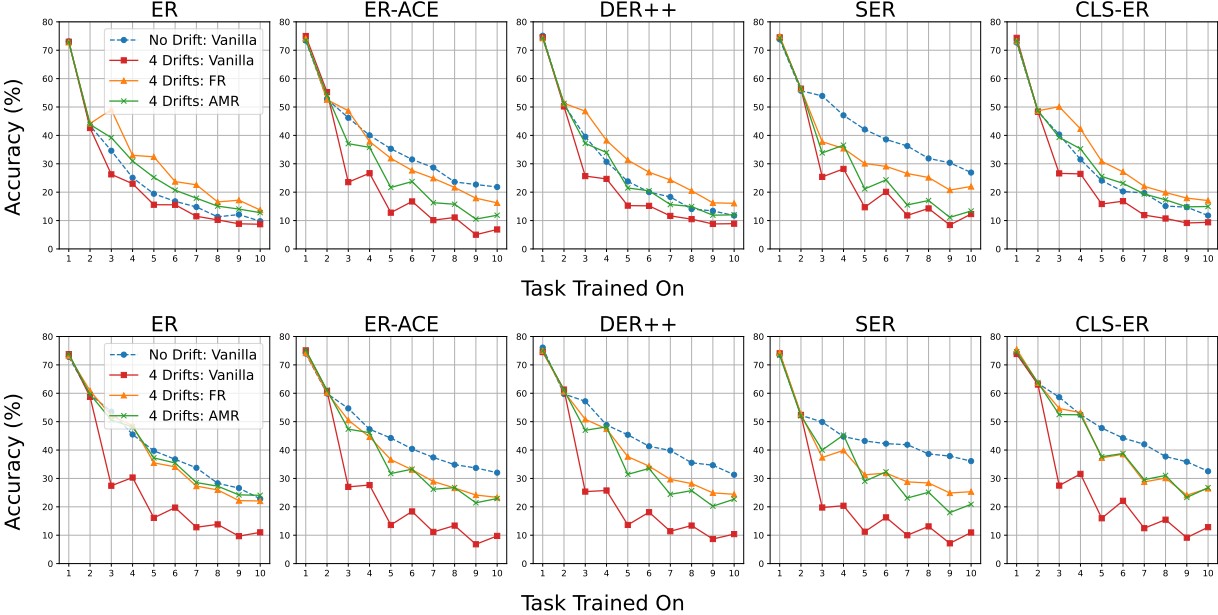

Figure 11: Class-incremental accuracy on S-CIFAR100 with drift events at tasks 3, 5, 7, and 9. Results are shown for buffer sizes 500 (top) and 5000 (bottom).

# 6    Conclusion, Limitations, and Future Works

**Conclusion:** We propose a novel continual learning scenario in which adaptation is required not only for newly arriving classes, but also for previously learned classes whose representations evolve over time due to concept drift. To address this setting, we introduce a holistic framework that couples drift detection with drift-aware memory management, allowing rehearsal-based learners to both retain stable knowledge and rapidly revise outdated concepts. Concretely, our Adaptive Memory Realignment (AMR) mechanism

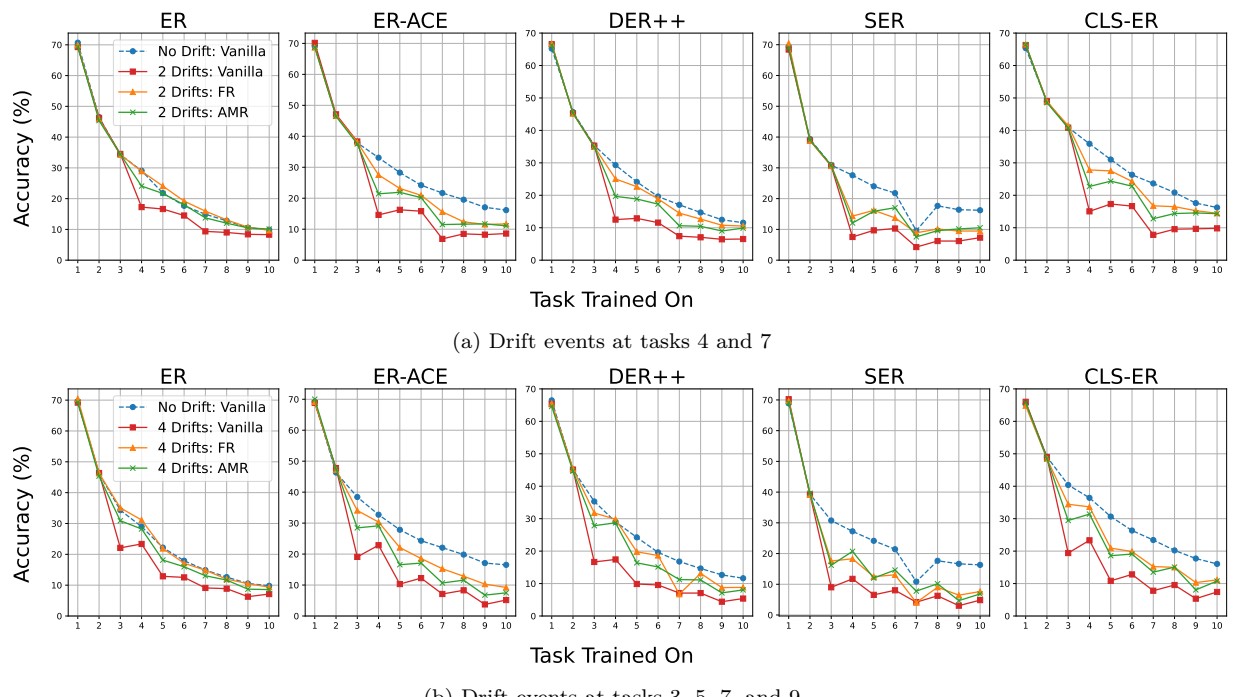

(a) Drift events at tasks 4 and 7

(b) Drift events at tasks 3, 5, 7, and 9

Figure 12: Class-incremental accuracy on S-Tiny-ImageNet-CD with 5000 buffer size.

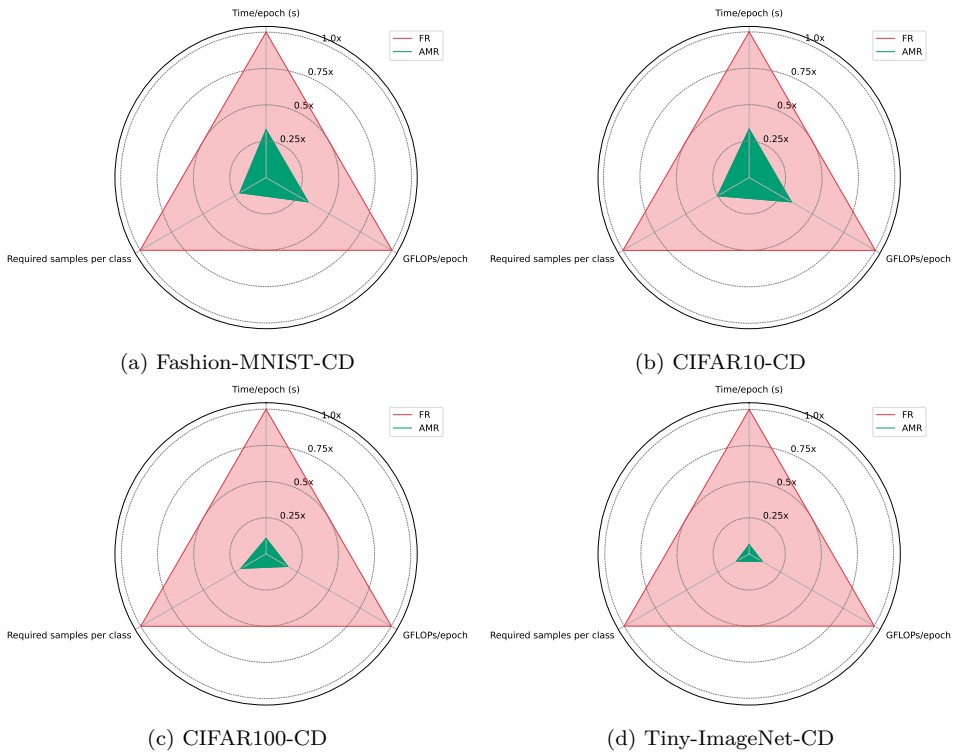

(a) Fashion-MNIST-CD

(b) CIFAR10-CD

(c) CIFAR100-CD

(d) Tiny-ImageNet-CD

Figure 13: Radar plots comparing the efficiency of AMR and FR across normalized metrics: time per epoch, GFLOPs, and labeled samples required. All values are normalized such that FR = 1.0 (maximum cost), with lower values indicating better efficiency. AMR consistently shows lower resource requirements across all dimensions.

selectively flushes stale buffer samples of drifted classes and repopulates those slots with a small number of up-to-date labeled instances, thereby realigning rehearsal gradients with the current data distribution. Across four drift-augmented vision benchmarks (Fashion-MNIST-CD, CIFAR10-CD, CIFAR100-CD, and Tiny-ImageNet-CD) and a real temporal-drift setting (CLEAR-10), AMR consistently recovers performance after drift while preserving accuracy on non-drifted tasks and does so with dramatically lower labeling and computational costs than full relearning from scratch. These findings highlight that "remembering" in continual learning is insufficient in non-stationary environments: robust agents must also detect when stored knowledge becomes stale and selectively update it. More broadly, by providing both a reproducible evaluation framework and strong empirical evidence for lightweight memory realignment, our work helps bridge continual learning and data-stream mining perspectives and encourages future research on end-to-end systems that jointly detect, diagnose, and adapt to diverse drift regimes in long-running deployments.

**Limitations:** Our framework currently relies on an external drift detector to trigger adaptation; thus, its end-to-end robustness is bounded by the detector's accuracy, calibration, and assumptions under distribution shift. In particular, false negatives can delay adaptation, leading to prolonged performance drops, while false positives can induce unnecessary memory realignments and additional labeling/compute overhead. Moreover, our specific instantiation (uncertainty-distribution testing against a buffer-derived reference) inherits practical limitations: (i) the reference distribution can be noisy when the buffer is small, highly imbalanced across classes, or already partially contaminated by earlier undetected drift; and (ii) some shifts may not manifest as a clear change in predictive uncertainty, reducing detector sensitivity. Beyond detector dependency, AMR may exhibit degraded performance in challenging drift scenarios, such as:

- **Adversarial drift concealment,** where an adaptive adversary smooths or masks distribution shifts to delay detection and retain corrupted samples in the buffer.

- **Gradual or mixture drift,** where overlapping pre- and post-drift subdomains produce weak, intermittent, or oscillatory detection signals, causing delayed or excessive reactions.

- **Open-set recurrence / novel modalities,** where previously seen classes reappear with new internal modalities or semantically related variants; in such cases, AMR may misinterpret semantic novelty as distributional drift (or vice versa), and naive full replacement may discard still-relevant sub-modes.

Finally, our study focuses on rehearsal-based continual image classification; while AMR is model-agnostic once drift is flagged, we do not claim a jointly optimized detection-adaptation pipeline, nor do we comprehensively evaluate extensions to non-rehearsal paradigms or other problem types, where drift signals and memory semantics may be substantially different. We therefore view detector choice, calibration, and supervision constraints as key practical considerations that must be tailored to the target application.

**Future Works:** Several research directions emerge from this work. First, hybrid detection strategies that combine multiple drift signals (e.g., uncertainty-based tests, feature-space divergence measures, and reconstruction errors) could improve robustness across diverse drift regimes, reducing both false positives and false negatives. Second, semantic drift monitoring that tracks class-level feature evolution rather than relying solely on distributional tests could better distinguish meaningful representation shifts from minor perturbations, minimizing unnecessary memory updates. Third, exploring learned or self-tuning drift detectors that adapt their sensitivity based on observed stream characteristics represents a promising avenue for reducing manual threshold calibration. Fourth, alternative pre-training strategies such as self-supervised temporal contrastive learning could yield detector features better suited to capturing gradual distributional evolution compared to ImageNet-pretrained representations optimized for classification. Finally, extending AMR to handle open-set recurrence and novel subclasses, where previously seen classes reappear with new internal modes or semantically related but novel classes emerge, would enhance applicability to more complex real-world scenarios. Collectively, these directions aim to develop more autonomous, adaptive continual learning systems capable of robust long-term deployment in non-stationary environments.

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

# A    Justification for Concept Drift Transformation

Figure 14 shows the effects of various transformations on recurring classes in CIFAR10, using vanilla Experience Replay (ER) with a buffer size of 5000. Specifically, during tasks 2 and 4, previously learned classes reappear alongside new classes, with the recurring ones modified using the indicated transformations to induce concept drift. The figure shows that certain transformations have a more pronounced impact than others. Defocus blur and shot noise fail to produce meaningful representation shifts, even at their highest severity levels. In contrast, Gaussian noise, rotation, and permutation significantly degrade performance, indicating stronger representation drift.

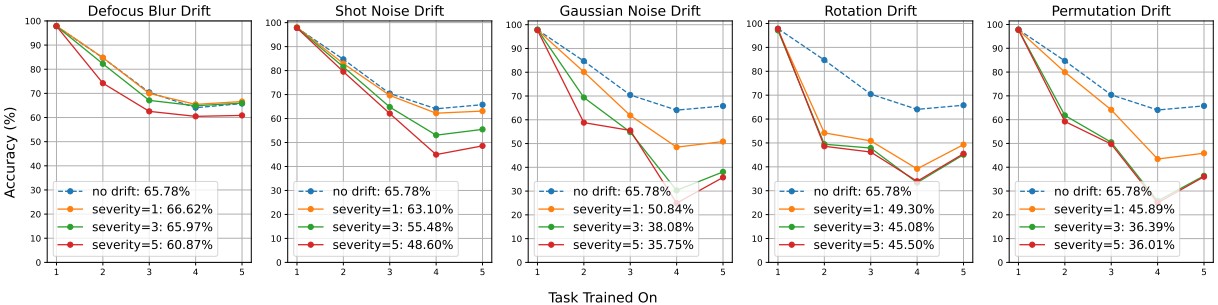

Figure 14: Impact of concept drifts induced by image transformations of varying severity at tasks 2 and 4 on CIFAR10

# B    Real-World Drift Experiments on CLEAR Dataset

To evaluate AMR's effectiveness on natural temporal distribution shifts beyond synthetic transformations, we conduct experiments on the CLEAR-10 dataset (Lin et al., 2021), which captures real-world visual concept evolution from 2004 to 2014.

**Experimental Setup:** We construct a 5-task class-incremental scenario using CLEAR-10, excluding the BACKGROUND class to maintain a 10-class setting. Each task introduces 2 classes drawn from temporal buckets 1–2 (approximately 2004–2005). Since each class contains only ∼300 training samples per bucket, we combine two consecutive buckets per class to increase sample counts. For non-drifted scenarios, training and test sets use the same temporal buckets to maintain distribution consistency. To induce drift, we replace the test distributions of recurring classes with samples from temporal buckets 9–10 (approximately 2012–2014), creating natural temporal shifts spanning 8–10 years. We evaluate ER, ER-ACE, and CLS-ER with a buffer size of $|\mathcal{M}| = 1000$ under a 2-drift scenario. DER++ and SER were excluded due to memory constraints on CLEAR-10.

**Drift Detection Challenges:** During implementation, we identified an important limitation of uncertainty-based drift detection on gradual temporal shifts. Our original KS-test on predictive uncertainty ($p = 0.05$) detected drift in only ∼50% of temporal shift scenarios. Relaxing the threshold to $p = 0.2$–$0.3$ yielded marginal improvement while compromising statistical significance. To address this, we integrated the Maximum Mean Discrepancy (MMD) drift detector (Gretton et al., 2012), which operates in feature space rather than output uncertainty space. MMD measures distribution distance in a Reproducing Kernel Hilbert Space by comparing expected feature embeddings between source and target distributions. This feature-level comparison proved significantly more sensitive to subtle visual evolution across temporal buckets, achieving near-perfect drift detection on CLEAR-10.

**Results and Discussion:** Figure 15 presents the results on Split-CLEAR10-CD. Unlike the synthetic drift benchmarks in the main paper, natural temporal shifts in CLEAR-10 produce subtle distribution changes. Vanilla adaptation shows minimal degradation between no-drift and 2-drift scenarios, indicating that 8–10 years of temporal evolution creates gentler drift than synthetic transformations such as permutation.

Table 5 shows that despite the subtle drift signal, AMR demonstrates consistent improvements on ER, achieving +7.17% FAA over Vanilla and +3.97% over FR. For CLS-ER, AMR slightly outperforms FR performance

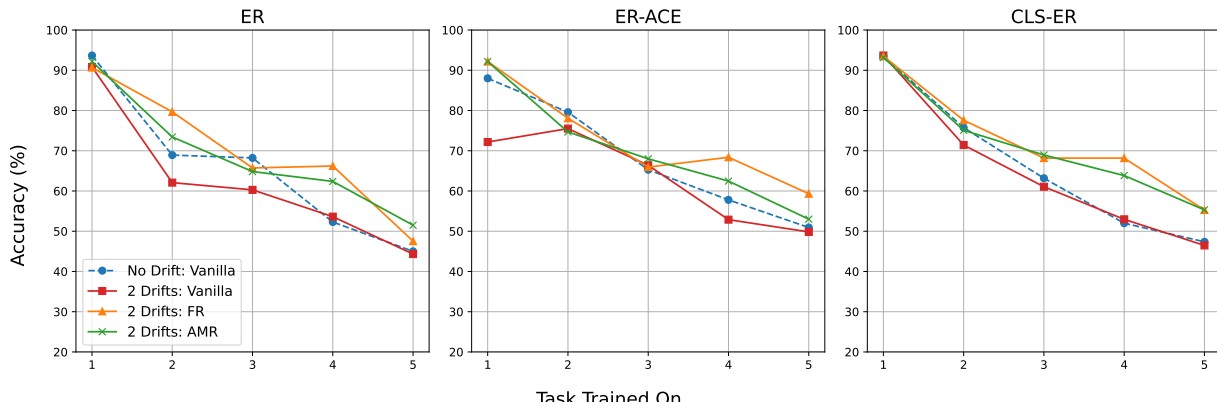

Figure 15: Class-incremental accuracy on Split-CLEAR10-CD with buffer size 1000. Drift events occur at tasks 2 and 4, where recurring classes are replaced with samples from temporal buckets 9-10 (2012∼2014). Natural temporal drift produces subtler distribution shifts compared to synthetic transformations, resulting in smaller performance gaps between adaptation strategies.

(55.30% vs. 55.27%). However, on ER-ACE, FR outperforms AMR (59.33% vs. 53.00%), suggesting that ER-ACE's asymmetric cross-entropy loss may interact differently with MMD-triggered realignment under subtle drift conditions.

Table 5: Final Average Accuracy (FAA[↑]) and Forgetting (F[↓]) for Split-CLEAR10-CD (3-run average).
*Vanilla = Baseline without Drift Adaptation, FR = Full Relearning, AMR = Adaptive Memory Realignment*

| $\mathcal{M}$ | Method | Adaptation (# drifts) | FAA$\uparrow_{\pm std}$ | F$\downarrow$ |
|---|---|---|---|---|
| 1000 | ER | Vanilla (0) | $45.00_{\pm 1.15}$ | 57.54 |
| | | Vanilla (2) | $44.33_{\pm 1.25}$ | 51.92 |
| | | FR (2) | $47.53_{\pm 0.77}$ | 52.92 |
| | | AMR (2) | $51.50_{\pm 2.05}$ | 50.33 |
| | ER-ACE | Vanilla (0) | $50.93_{\pm 1.19}$ | 29.92 |
| | | Vanilla (2) | $49.83_{\pm 1.14}$ | 28.00 |
| | | FR (2) | $59.33_{\pm 0.98}$ | 25.92 |
| | | AMR (2) | $53.00_{\pm 0.59}$ | 28.96 |
| | CLS-ER | Vanilla (0) | $47.37_{\pm 0.29}$ | 53.62 |
| | | Vanilla (2) | $46.47_{\pm 0.74}$ | 55.12 |
| | | FR (2) | $55.27_{\pm 3.00}$ | 38.00 |
| | | AMR (2) | $55.30_{\pm 3.13}$ | 42.92 |

These results reveal important insights about the relationship between drift magnitude and adaptation strategy effectiveness. While AMR excels at recovering from pronounced synthetic drifts, its advantages are attenuated when drift signals are subtle and gradual. This aligns with the limitations discussed in Section 6: gradual drift produces weak detection signals that can cause suboptimal adaptation timing. Nevertheless, AMR remains competitive with or superior to FR across most settings while requiring substantially fewer labeled samples, confirming its utility even in challenging real-world drift scenarios.

## C  Backbone Justification and Robustness Across Architectures

To validate that AMR's effectiveness generalizes beyond ResNet-18, we conducted additional experiments using deeper convolutional (ResNet-152) and transformer-based (ViT-S) architectures. We evaluate Ex-

perience Replay (ER) on S-CIFAR10-CD with 2 drifts and S-CIFAR100-CD with 4 drifts under identical conditions (permutation drift, buffer size $|\mathcal{M}| = 5000$).

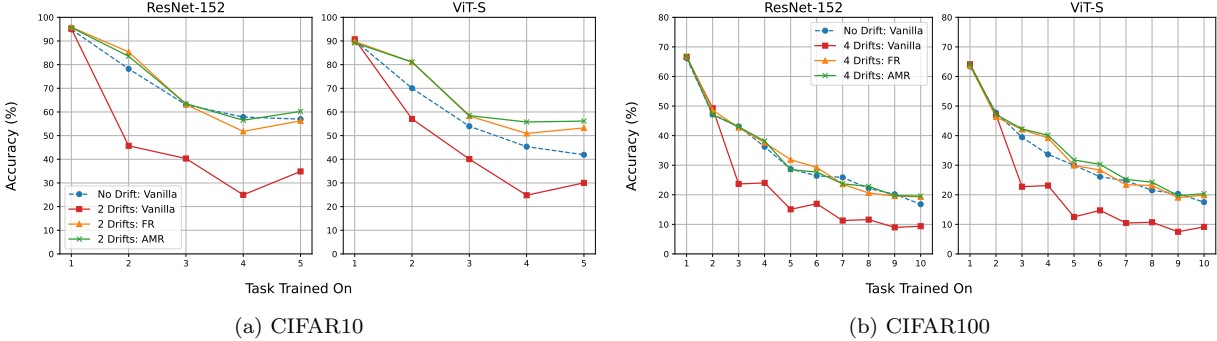

(a) CIFAR10                                   (b) CIFAR100

Figure 16: Class-incremental accuracy using ResNet-152 and ViT-S backbones with Experience Replay (ER) and buffer size 5000. Drift events occur at tasks 2 and 4 for CIFAR10, and tasks 3, 5, 7, and 9 for CIFAR100.

Table 6: Final Average Accuracy (FAA[↑]) and Forgetting (F[↓]) across different backbone architectures.

| Dataset | Backbone | Adaptation (# drifts) | FAA↑ | F↓ |
|---|---|---|---|---|
| S-CIFAR10-CD | ResNet-152 | Vanilla (0) | 56.97 | 41.04 |
| | | Vanilla (2) | 34.87 | 69.45 |
| | | FR (2) | 56.26 | 42.20 |
| | | AMR (2) | 60.23 | 38.30 |
| | ViT-S | Vanilla (0) | 41.91 | 50.54 |
| | | Vanilla (2) | 30.04 | 65.21 |
| | | FR (2) | 53.26 | 37.14 |
| | | AMR (2) | 56.15 | 35.35 |
| S-CIFAR100-CD | ResNet-152 | Vanilla (0) | 16.80 | 52.17 |
| | | Vanilla (4) | 9.37 | 60.71 |
| | | FR (4) | 19.28 | 48.98 |
| | | AMR (4) | 19.56 | 50.53 |
| | ViT-S | Vanilla (0) | 17.53 | 45.94 |
| | | Vanilla (4) | 9.15 | 54.69 |
| | | FR (4) | 19.86 | 44.14 |
| | | AMR (4) | 20.37 | 47.48 |

As shown in Table 6 and Figure 16, AMR consistently outperforms both the Vanilla baseline and Full Re-learning (FR) across both convolutional and transformer architectures. On S-CIFAR10-CD, AMR achieves gains of +3.97% FAA over FR with ResNet-152 and +2.89% with ViT-S. The performance trends observed with ResNet-18 in the main experiments are preserved across architectures, confirming that AMR is architecture-agnostic. The choice of ResNet-18 for the main experiments was motivated by computational efficiency, enabling comprehensive evaluation across multiple datasets, buffer sizes, and drift scenarios while maintaining tractable training times.

## D  Hyperparameter Selection

Abbreviations: $mb$ = mini-batch size, $bs$ = batch size, $reg_w$ = regularization weight, $sm\_uf$ = stable model update frequency, $pm\_uf$ = plastic model update frequency

Table 7: Hyperparameters for S-FMNIST-CD and S-CIFAR10-CD.

| Method | $\mathcal{M}$ | Hyperparameters | |
| --- | --- | --- | --- |
| | | *S-FMNIST-CD* | *S-CIFAR10-CD* |
| ER | 500 | $lr$: 0.1, $mb$: 10, $bs$: 10, $epochs$: 1 | $lr$: 0.1, $mb$: 32, $bs$: 32, $epochs$: 50 |
| | 5000 | $lr$: 0.1, $mb$: 10, $bs$: 10, $epochs$: 1 | $lr$: 0.1, $mb$: 32, $bs$: 32, $epochs$: 50 |
| ER-ACE | 500 | $lr$: 0.03, $mb$: 10, $bs$: 10, $epochs$: 1 | $lr$: 0.03, $mb$: 32, $bs$: 32, $epochs$: 50 |
| | 5000 | $lr$: 0.03, $mb$: 10, $bs$: 10, $epochs$: 1 | $lr$: 0.03, $mb$: 32, $bs$: 32, $epochs$: 50 |
| DER++ | 500 | $lr$: 0.1, $mb$: 10, $bs$: 10, $\alpha$: 0.2, $\beta$: 0.5, $epochs$: 1 | $lr$: 0.03, $mb$: 32, $bs$: 32, $\alpha$: 0.2, $\beta$: 0.5, $epochs$: 50 |
| | 5000 | $lr$: 0.1, $mb$: 10, $bs$: 10, $\alpha$: 0.2, $\beta$: 0.5, $epochs$: 1 | $lr$: 0.03, $mb$: 32, $bs$: 32, $\alpha$: 0.1, $\beta$: 1.0, $epochs$: 50 |
| SER | 500 | $lr$: 0.1, $mb$: 10, $bs$: 10, $\alpha$: 0.2, $\beta$: 0.2, $epochs$: 1 | $lr$: 0.03, $mb$: 32, $bs$: 32, $\alpha$: 0.2, $\beta$: 0.2, $epochs$: 50 |
| | 5000 | $lr$: 0.1, $mb$: 10, $bs$: 10, $\alpha$: 0.2, $\beta$: 0.2, $epochs$: 1 | $lr$: 0.03, $mb$: 32, $bs$: 32, $\alpha$: 0.2, $\beta$: 0.2, $epochs$: 50 |
| CLS-ER | 500 | $lr$: 0.1, $mb$: 10, $bs$: 10, $reg\_w$: 1.0, $sm\_uf$: 0.9, $sm_\alpha$: 0.99, $pm\_uf$: 1.0, $pm_\alpha$: 0.99, $epochs$: 1 | $lr$: 0.1, $mb$: 32, $bs$: 32, $reg\_w$: 0.15, $sm\_uf$: 0.1, $sm_\alpha$: 0.999, $pm\_uf$: 0.9, $pm_\alpha$: 0.999, $epochs$: 50 |
| | 5000 | $lr$: 0.1, $mb$: 10, $bs$: 10, $reg\_w$: 1.0, $sm\_uf$: 0.8, $sm_\alpha$: 0.99, $pm\_uf$: 1.0, $pm_\alpha$: 0.99, $epochs$: 1 | $lr$: 0.1, $mb$: 32, $bs$: 32, $reg\_w$: 0.15, $sm\_uf$: 0.8, $sm_\alpha$: 0.999, $pm\_uf$: 1.0, $pm_\alpha$: 0.999, $epochs$: 50 |

Table 8: Hyperparameters for S-CIFAR100-CD and S-Tiny-ImageNet-CD.

| Method | $\mathcal{M}$ | Hyperparameters | |
| --- | --- | --- | --- |
| | | *S-CIFAR100-CD* | *S-Tiny-ImageNet-CD* |
| ER | 500 | $lr$: 0.1, $mb$: 32, $bs$: 32, $epochs$: 50 | $lr$: 0.03, $mb$: 32, $bs$: 32, $epochs$: 100 |
| | 5000 | $lr$: 0.1, $mb$: 32, $bs$: 32, $epochs$: 50 | $lr$: 0.1, $mb$: 32, $bs$: 32, $epochs$: 100 |
| ER-ACE | 500 | $lr$: 0.03, $mb$: 32, $bs$: 32, $epochs$: 50 | $lr$: 0.03, $mb$: 32, $bs$: 32, $epochs$: 100 |
| | 5000 | $lr$: 0.03, $mb$: 32, $bs$: 32, $epochs$: 50 | $lr$: 0.03, $mb$: 32, $bs$: 32, $epochs$: 100 |
| DER++ | 500 | $lr$: 0.03, $mb$: 32, $bs$: 32, $\alpha$: 0.1, $\beta$: 0.5, $epochs$: 50 | $lr$: 0.03, $mb$: 32, $bs$: 32, $\alpha$: 0.2, $\beta$: 0.5, $epochs$: 100 |
| | 5000 | $lr$: 0.03, $mb$: 32, $bs$: 32, $\alpha$: 0.1, $\beta$: 0.5, $epochs$: 50 | $lr$: 0.03, $mb$: 32, $bs$: 32, $\alpha$: 0.1, $\beta$: 0.5, $epochs$: 100 |
| SER | 500 | $lr$: 0.03, $mb$: 32, $bs$: 32, $\alpha$: 0.5, $\beta$: 0.5, $epochs$: 50 | $lr$: 0.03, $mb$: 32, $bs$: 32, $\alpha$: 0.2, $\beta$: 1.0, $epochs$: 100 |
| | 5000 | $lr$: 0.03, $mb$: 32, $bs$: 32, $\alpha$: 0.5, $\beta$: 0.5, $epochs$: 50 | $lr$: 0.03, $mb$: 32, $bs$: 32, $\alpha$: 0.2, $\beta$: 1.0, $epochs$: 100 |
| CLS-ER | 500 | $lr$: 0.1, $mb$: 32, $bs$: 32, $reg\_w$: 0.15, $sm\_uf$: 0.1, $sm_\alpha$: 0.999, $pm\_uf$: 0.9, $pm_\alpha$: 0.999, $epochs$: 50 | $lr$: 0.05, $mb$: 32, $bs$: 32, $reg\_w$: 0.1, $sm\_uf$: 0.05, $sm_\alpha$: 0.999, $pm\_uf$: 0.08, $pm_\alpha$: 0.999, $epochs$: 100 |
| | 5000 | $lr$: 0.1, $mb$: 32, $bs$: 32, $reg\_w$: 0.15, $sm\_uf$: 0.8, $sm_\alpha$: 0.999, $pm\_uf$: 1.0, $pm_\alpha$: 0.999, $epochs$: 50 | $lr$: 0.05, $mb$: 32, $bs$: 32, $reg\_w$: 0.1, $sm\_uf$: 0.07, $sm_\alpha$: 0.999, $pm\_uf$: 0.08, $pm_\alpha$: 0.999, $epochs$: 100 |

