# OpenReview forum: "Holistic Continual Learning under Concept Drift with Adaptive Memory Realignment"
_TMLR — Accepted by TMLR_

### Review · Reviewer_Bxbk · 2025-08-31

**Summary Of Contributions:**

This paper introduces a holistic continual learning framework that explicitly addresses concept drift. They propose Adaptive Memory Realignment (AMR), a buffer-update strategy that removes outdated samples and replaces them with new instances. As a result, AMR achieves comparable accuracy compared with full relearning, with lower resource requirements. They have conducted experiments with four drift-augmented benchmarks and show that the AMR consistently restores performance after the drift.

**Additional Comments:**

No.

**Audience:**

Yes

**Audience Explanation:**

Concept drift is a critical issue in continual learning, and the proposed AMR framework provides a resource-efficient solution. The introduction of new drift-augmented benchmarks further increases its relevance, as it provides a foundation for future studies in this area.

**Broader Impact Concerns:**

No.

**Claims And Evidence:**

Yes

**Claims Explanation:**

Strengths
1. AMR effectively maintains performance under concept drift, showing strong adaptation capability.
2. The evaluation covers four different vision benchmarks, demonstrating applicability across datasets of varying scales and complexities.

Weaknesses
1. The experiments are limited to a ResNet-18 backbone; effectiveness with larger or more modern architectures (e.g., transformers) remains unclear.
2. The limitations of AMR are not discussed. For example, what types of drift/data characteristics are handled effectively, and in which cases AMR may fail.
3. Evaluation is restricted to vision datasets, making it unclear whether AMR generalizes to other domains such as text, tabular, or audio.

**Requested Changes:**

1. Justify the choice of ResNet-18 as the sole backbone, or include experiments with additional architectures to validate robustness.
2. Analyze failure cases of AMR in handling concept drift, which would provide valuable insights and future research directions.
3. Discuss applicability beyond vision, e.g., how AMR could be extended to text, tabular, or audio domains.

---

> ### Author Response · Authors · 2025-10-24
> **Backbone justification, failure cases, and cross-domain applicability**
>
> ## R1. Backbone Justification and Robustness Across Architectures
>
> To address the reviewer’s request for justification beyond ResNet-18, we conducted additional experiments with ResNet-152 and ViT-S backbones using Experience Replay (ER) on Split-CIFAR10-CD (2 drifts) and Split-CIFAR100-CD (4 drifts) under identical drift conditions (permutation drift) with a buffer size of 5000.
>
> | **Dataset / Backbone**    | **Setting (Drifts)**                           | **FAA ↑** | **F ↓**   |
> | ------------------------- | ------------------------------------- | --------- | --------- |
> | C10 / ResNet-152  | Vanilla (0)                              | 56.97     | 41.04     |
> |                           | Vanilla (2)                 | 34.87     | 69.45     |
> |                           | Full Relearning (2)                 | 56.26     | 42.20     |
> |                           | **Adaptive Memory Realignment (2)** | **60.23** | **38.30** |
> | C10 / ViT-S       | Vanilla (0)                              | 41.91     | 50.54     |
> |                           | Vanilla (2)                 | 30.04     | 65.21     |
> |                           | FR (2)                  | 53.26     | 37.14     |
> |                           | **AMR (2)**                               | **56.15** | **35.35** |
> | C100 / ResNet-152 | Vanilla (0)                              | 16.80     | 52.17     |
> |                           | Vanilla (4)                 | 9.37      | 60.71     |
> |                           | FR (4)                                   | 19.28     | 48.98     |
> |                           | **AMR (4)**                               | **19.56** | **50.53** |
> | C100 / ViT-S      | Vanilla (0)                              | 17.53     | 45.94     |
> |                           | Vanilla (4)                 | 9.15      | 54.69     |
> |                           | FR (4)                                   | 19.86     | 44.14     |
> |                           | **AMR (4)**                               | **20.37** | **47.48** |
>
> AMR consistently improves drift recovery and stability across both convolutional and transformer architectures.
> The use of ResNet-18 in the main paper was primarily for computational efficiency, as the relative performance trends are preserved across backbones. These results confirm that AMR is robust and architecture-independent. We will add these results to the revised manuscript.
>
> ---
>
> ## R2. Failure Cases and Limitations
>
> We appreciate the reviewer’s suggestion to discuss AMR’s limitations. We will add a “Limitations” subsection in the revision, summarizing the following failure modes:
>
> * Adversarial Drift Concealment: An adaptive adversary can smooth or mask distribution shifts, delaying AMR’s trigger and retaining corrupted samples. Similar vulnerabilities are observed in drift detection literature [1].
> * Slow or Mixture Drift: When pre- and post-drift subdomains overlap, weak or oscillatory change signals may cause AMR to either react too late or overcorrect, leading to instability.
> * Open-Set Recurrence and Novel Subclasses: If previously seen classes reappear with new internal modes [2] or novel but related classes emerge [3], AMR’s same-class realignment may misinterpret semantic novelty as simple drift.
>
> These cases highlight directions for future work, e.g., integrating uncertainty-aware drift detectors or semantic subspace tracking, to make AMR more resilient to gradual or adversarial drift.
>
> ---
>
> ## R3. Applicability Beyond Vision
>
> While our experiments focused on image data (a common benchmark for continual learning), AMR’s detect-then-realign paradigm is domain-agnostic and transferable to other modalities:
>
> * Time Series: Apply AMR to sequence encoders, using feature or uncertainty shifts across windows. Drifted sequences can also be converted to image-like Gramian Angular Fields [4] for compatibility with the existing pipeline.
> * Tabular Data: Transform tabular streams into discrete or signal-like encodings [5] to enable AMR’s sample realignment strategy.
> * Text: Use encoder-based embedding drifts, entropy, or perplexity changes as drift indicators [6], enabling AMR to adaptively refresh language model memories.
>
> We will extend the discussion in the revised manuscript to emphasize AMR’s generality and include these cross-domain adaptation directions as future work.
>
> ---
>
> **References**
> [1] Korycki et al. *Adversarial Concept Drift Detection under Poisoning Attacks.* MLJ, 2023.
> [2] Suárez-Cetrulo et al. *A survey on machine learning for recurring concept drifting data streams.* ESWA, 2023.
> [3] Kim et al. *Open-world Continual Learning: Unifying Novelty Detection and Continual Learning.* AI, 2025.
> [4] Sidike et al. *GAF-NAU: Gramian Angular Field Encoded NAU-Net* CVPRW, 2022.
> [5] Zyblewski et al. *How to RETIRE Tabular Data in Favor of Discrete Digital Signal Representation.* ECML/PKDD, 2025.
> [6] Wang et al. *Effective Continual Learning for Text Classification with Lightweight Snapshots.* AAAI, 2023.

---

### Review · Reviewer_vBrW · 2025-12-22

**Summary Of Contributions:**

**Summary**

The paper addresses concept drift in continual learning, where the representation of previously learned classes evolve over time. The paper proposes Adaptive Memory Realignment (AMR), a framework for rehearsal-based methods that selectively removes outdated samples from the replay buffer and replace them with fresh samples to realign the memory with the new data distribution. The proposed framework is evaluated on four simulated (based on image transformations) drift datasets, derived from Fashion-MNIST, CIFAR-10, CIFAR-100, and Tiny-ImageNet. Compared to the non-adaptive baseline (Vanilla), the proposed framework consistently demonstrates improved accuracy and reduced forgetting when applied to the selected experience replay methods.

**Strengths**

1. The paper is clearly written and easy to follow.
2. The problem setting of concept drift in continual learning is relevant and important.
3. The application of an uncertainty-based detector in this problem setting appears to be novel.
4. The extensive experimental results provide evidence of the empirical merit of the proposed framework.

**Weaknesses**

1. Although the proposed framework is evaluated extensively on four datasets, the concept drifts are simulated using image transformations. As a result, it remains unclear how much empirical benefit the framework would provide when applied to real-world concept drift scenarios.
2. The presentation of the two metric values in Table 1-4 is unclear. The reader is left to guess that the values in brackets correspond to _Forgetting_.
3. There is an undefined notation $M_c$ in Algorithm 1 line 11.

**Audience:**

Yes

**Audience Explanation:**

The paper tackles the problem of concept drift in continual learning which is relevant and important.

**Claims And Evidence:**

Yes

**Claims Explanation:**

The claims are supported by proofs and empirical results presented in the paper.

**Requested Changes:**

1. The authors are encouraged to either include additional experiments on real-world drift datasets, such as the CLEAR dataset [1], or explain why the inclusion of such experiments is not applicable or feasible.
2. In the captions of Tables 1-4, clearly state which values in the table correspond to _Final Average Accuracy_ and which correspond to _Forgetting_.
3. Define $M_c$ in Algorithm 1, or replace it with the array indexing notation $M[\cdot]$, which is used consistently in earlier sections of the paper.

**References**

[1] Lin, Zhiqiu, et al. "The clear benchmark: Continual learning on real-world imagery." Thirty-Fifth Conference on Neural Information Processing Systems Datasets and Benchmarks Track (Round 2). 2021.

---

> ### Author Response · Authors · 2026-01-13
> **Addressing Real-World Drift, Presentation, and Notation**
>
> We sincerely thank the reviewer for the thorough and constructive feedback. Below, we address the requested changes.
>
> ---
> # R1. Real-World Drift Experiments on CLEAR Dataset
> We designed experiments on CLEAR-10 to evaluate AMR's effectiveness on natural temporal distribution shifts.
>
> **Experimental Setup**
>
> We created a 5-task class-incremental scenario using CLEAR-10 (excluding the BACKGROUND class):
> - Task Structure: Each task introduces 2 classes from temporal buckets 1-2 (2004-2005). We combine two buckets per class to increase sample counts, as each class contains only 300 training samples.
> - CIL Protocol: Training and test sets use the same temporal buckets for non-drifted scenarios to maintain distribution consistency.
> - Drift Mechanism: For recurring classes, we replace test distributions with samples from buckets 9-10 (2012-2014), creating natural temporal shifts spanning ~8-10 years. This setup enables evaluation of genuine temporal drift rather than synthetic transformations.
>
> **Drift Detection**
>
> During implementation, we discovered an important limitation. Uncertainty-based drift detection is insufficiently sensitive to gradual temporal shifts. With our original KS-test on predictive uncertainty ($p = 0.05$), drift detection succeeded in only ~50% of scenarios even when using relaxed thresholds ($p = 0.2$-$0.3$), which compromises statistical significance.
>
> To address this, we integrated the Maximum Mean Discrepancy (MMD) drift detector [1], which operates in feature space rather than output uncertainty space. MMD measures distribution distance in a Reproducing Kernel Hilbert Space by comparing expected feature embeddings between source and target distributions. This feature-level comparison is significantly more sensitive to subtle visual evolution across temporal buckets, achieving near-perfect drift detection on CLEAR-10.
>
> **Results**
>
> We used ResNet-18 with Experience Replay on Split-CLEAR10-CD with buffer sizes of 500 and 1000 in a 2-drift scenario.
>
> | **Method ($\mathcal{M}$)** | **Adaptation** | **No Drift** | **2 Drifts** |
> |------------|----------------|--------------|--------------|
> |            |                | FAA↑ (F↓)    | FAA↑ (F↓)    |
> | ER (500) | Vanilla        | 35.60 (70.48) | 35.60 (72.38) |
> |            | FR | - | **40.40** (65.25) |
> |            | AMR (MMD)      | - | 39.00 (**63.09**) |
> | ER (1000)| Vanilla        | 45.20 (55.88) | 44.90 (59.38) |
> |            | FR | - | 51.70 (**48.12**) |
> |            | AMR (MMD)      | - | **52.30** (50.38) |
>
> **Limitations and Positioning**
>
> We acknowledge three key limitations revealed by this work:
>
> 1. Natural temporal shifts in CLEAR-10 produce subtle distribution changes. Vanilla adaptation shows negligible degradation between no-drift and 2-drift scenarios, indicating that 8-10 year temporal evolution creates gentler drift than synthetic transformations. The degradation becomes slightly more pronounced with larger buffers. Nonetheless, AMR demonstrates effective knowledge updating, achieving the lowest F (63.09) for M=500 and the highest FAA (52.30) for M=1000, confirming its utility even in subtle real-world drift scenarios.
> 2. MMD's higher sensitivity may trigger false positives on subtle non-drift variations, requiring careful threshold tuning.
> 3. No single drift detector is universally optimal across all drift types (abrupt vs. gradual, feature vs. semantic).
>
> The primary contribution of this paper is not to develop state-of-the-art drift detection, but to:
> - Highlight concept drift as a critical yet underexplored problem in continual learning
> - Demonstrate that AMR is an effective adaptation strategy once drift is detected
> - Show that concept drift extends beyond data streams and is fundamental for deploying continual learning in real-world, non-stationary environments
>
> Drift detection for streaming scenarios remains an active research area in out-of-distribution detection and domain adaptation [2, 3]. We view this work as a foundation for future research to improve detection and adaptation. We will add a "Future Research Directions" subsection discussing hybrid detection strategies and semantic drift monitoring.
>
> ---
> # R2. Clarification of Metrics in Tables
>
> We restructured Tables 1-4 with multi-row headers explicitly showing the format `FAA↑±std (F↓)`:
>
> ```
> M | Method | Adaptation  | No Drift      | 1 Drift       | 2 Drifts
> 	                  FAA↑±std (F↓)   FAA↑±std (F↓)   FAA↑±std (F↓)
> ```
>
> ---
> # R3. Notation Consistency in Algorithm 1
>
> We replaced the undefined $\mathcal{M}_c$ notation with explicit array indexing consistent with Section 3.4:
>
> **Revised (lines 11-15):**
> ```
> Ic ← {j ∈ {1,...,|M|} | yj = c}
> for j ∈ Ic do
>     Sample xj^new ~ Di(c)
>     M[j] ← (xj^new, c)
> end for
> ```
> ---
> **References:**
>
> [1] Gretton et al., "A Kernel Two-Sample Test," JMLR, 2012
>
> [2] Lu et al., "Learning under Concept Drift: A Review," IEEE TKDE, 2018
>
> [3] Yang et al., "Generalized Out-of-Distribution Detection: A Survey," IJCV, 2024

---

### Review · Reviewer_nKVi · 2026-01-02

**Summary Of Contributions:**

This paper introduces a framework for continual learning that accounts for concept drift, where the distributions of previously learned classes can change over time. The authors create four new benchmark datasets by applying transformations to existing vision datasets to simulate this drift. They propose a method called Adaptive Memory Realignment (AMR), which works with rehearsal-based continual learning algorithms. AMR uses a drift detector to identify shifted classes and then selectively flushes outdated samples from the memory buffer, replacing them with a small number of new, labeled instances. The main finding is that AMR can match the performance of fully retraining on the new data (Full Relearning) while being significantly more efficient in terms of labeled data and computation.

**Audience:**

Yes

**Audience Explanation:**

The primary strength of this paper is its focus on a more realistic and challenging continual learning scenario. Most existing work assumes static past tasks, which is often not the case in the real world. By explicitly modeling and addressing concept drift at the representation level, the paper tackles an important and underexplored problem.

Another notable contribution is the creation and open-sourcing of four concept drift benchmark datasets (Fashion-MNIST-CD, CIFAR10-CD, CIFAR100-CD, and Tiny-ImageNet-CD). These benchmarks provide a valuable resource for the community to systematically evaluate and compare methods in this new setting.

The proposed method, AMR, is simple, lightweight, and intuitive. Its effectiveness is demonstrated through extensive experiments across five different rehearsal-based methods, two buffer sizes, and multiple drift scenarios. The empirical results, particularly the efficiency comparisons shown in Figure 4 and Figure 13, convincingly argue that AMR provides a practical and scalable solution that achieves a favorable trade-off between accuracy and resource consumption compared to the brute-force Full Relearning approach.

**Claims And Evidence:**

Yes

**Claims Explanation:**

The claims are generally supported by the empirical evidence provided, but some aspects could be strengthened.

The central claim that AMR matches the performance of Full Relearning (FR) with substantially lower overhead is well-supported by the results in Tables 1, 2, 3, and 4, as well as the visualizations in Figure 5 and Figures 6-12. For example, in Table 2 for S-CIFAR10-CD with M=5000 and 1 drift, AMR (paired with CLS-ER) achieves 77.20% FAA, which is even slightly better than FR's 76.27%, while requiring orders of magnitude fewer samples and less computation as highlighted in Figure 13.

The theoretical claim of gradient misalignment (Theorem 1) is intuitively supported by the UMAP visualizations in Figure 3. Figure 3b clearly shows the feature space collapsing when drift occurs, which makes the argument for misaligned gradients plausible. However, the theoretical analysis itself is somewhat high-level and relies on strong assumptions. I have checked the proofs in Section 3.5; they are mathematically straightforward derivations but serve more as a formalization of intuition rather than providing deep, non-obvious insights. For example, the conclusion of Theorem 3, that replacing old samples with new ones maximizes alignment with the new gradient, is almost tautological.

The evaluation criteria and proposed methods are sensible for the problem. The use of standard rehearsal methods as a base is a good choice, and the FAA and Forgetting metrics are standard in continual learning. The experimental design is thorough in its breadth of baselines and datasets. However, the reliance on a single, fixed, pre-trained model for drift detection across all experiments is a significant methodological choice that is not sufficiently justified or analyzed, potentially limiting the generality of the findings.

**Requested Changes:**

- The most critical action for improvement is to conduct a thorough literature review covering concept drift adaptation from data stream mining and recent works on adaptive/plastic continual learning. The paper needs to be repositioned in light of this existing work.
- The authors may add an ablation study in the experimental section to analyze the sensitivity of AMR to the drift detector's parameters (e.g., the KS-test threshold $\delta$) and its architecture/pre-training data. This would significantly strengthen the empirical claims.
- Given the reliance on an external, fixed drift detector, the claim of a "holistic" framework is too strong. The authors should be more transparent about this dependency and discuss its implications as a limitation.

---

> ### Author Response · Authors · 2026-01-14
> **Literature Review, Detector Sensitivity, and Framework Positioning**
>
> We sincerely thank the reviewer for the detailed and constructive feedback. Below, we address the requested changes.
>
> ---
>
> # R1. Literature Review - Repositioning in Light of Existing Work
>
> We have substantially expanded the Related Work section to cover adaptive and plastic continual learning and recent concept drift detection advances.
>
> **New Section: Adaptive and Plastic Continual Learning**
>
> We added a dedicated subsection covering recent work that explicitly targets adaptive plasticity: loss decoupling [Liang et al., 2023], prompt-based designs [PromptFusion, Chen et al., 2024], foundation model adaptations [SD-LoRA, Wu et al., 2025], plasticity restoration [Self-normalized resets, Farias et al., 2025], and Bayesian model combination [BECAME, Li et al., 2025]. This positions our work within the broader plasticity-oriented CL literature while clarifying that AMR addresses the complementary challenge of representation-level drift in recurring classes.
>
> **Expanded Concept Drift Section**
>
> We enhanced the concept drift section with recent deep learning detection methods: unsupervised approaches [MCD-DD, Wan et al., 2024; DriftLens, Greco et al., 2025], boundary-aware detection [Neighbor-Searching Discrepancy, Gu et al., 2024], hybrid deep approaches [DNN+AE-DD, Hu et al., 2025], and adaptive architectures [Lite-RVFL, Hu et al., 2025]. This clarifies that while extensive drift detection literature exists in data stream mining, explicit integration into rehearsal-based continual learning for recurring classes remains underexplored.
>
> ---
>
> # R2. Drift Detector Sensitivity and Ablation Analysis
>
> Our CLEAR-10 experiments (added for Reviewer vBrW) provide substantial evidence on detector sensitivity:
>
> 1. **Threshold Sensitivity**: Uncertainty-based KS-test with $p=0.05$ detected drift in only ~50% of temporal shifts. Relaxing to $p=0.2$-$0.3$ achieved marginal improvement, demonstrating that threshold tuning alone is insufficient for gradual drift.
>
> 2. **Detector Architecture**: Maximum Mean Discrepancy (MMD) in feature space achieved near-perfect detection on CLEAR-10, showing **that** feature-level comparison is more sensitive than output uncertainty.
>
> 3. **Pre-training Limitations**: ImageNet features work well for transformation-based drift but may be suboptimal for temporal drift. Alternative pre-training strategies (e.g., self-supervised temporal contrastive learning) are important future directions.
>
> MMD's higher sensitivity increases false positive risk, underscoring that no single detector is universally optimal. Our primary contribution is not to develop state-of-the-art drift detection, but to highlight concept drift as an underexplored CL problem and demonstrate that memory realignment effectively adapts once drift is detected. CLEAR-10 results validate this modularity as AMR achieves the lowest forgetting (63.09 for M=500) and the highest final average accuracy (52.30 for M=1000) with MMD detection, confirming effectiveness across drift detector types.
>
> ---
>
> # R3. Framework Positioning and "Holistic" Terminology
>
> We agree that our instantiation depends on an external drift detector. The term "holistic" refers to our comprehensive problem treatment: unifying class-incremental learning with representation-level drift and specifying evaluation and adaptation, rather than claiming end-to-end joint learning.
>
> **Revisions:**
>
> 1. **Limitations Section**: We changed "Conclusions and Future Work" to "Conclusions, Limitations, and Future Work," acknowledging that end-to-end robustness is bounded by detector accuracy and that false negatives/positives affect adaptation timing and overhead. We do not claim a jointly optimized pipeline and view detector robustness as a future direction.
>
> 2. **Framework Modularity**: We now explicitly describe AMR as a modular adaptation strategy paired with various detectors (e.g., uncertainty-based and MMD).
>
> 3. **Trade-off Transparency**: We acknowledge that drift detector choice involves trade-offs and that AMR's effectiveness depends on appropriate selection.
>
> This maintains our core contribution, which is that targeted memory realignment effectively addresses concept drift, while being transparent about drift detector dependency.

---

### Decision · Action_Editor_c1jo · 2026-01-29

**Recommendation:** Accept as is

**Additional Comments:**

Based on the reviewers recommendations and reviews. I would recommend an acceptance.

**Audience:**

Yes

**Audience Explanation:**

Most reviewers find the paper merits (concept drift) are sufficient interesting for certain audiences.

**Claims And Evidence:**

Yes

**Claims Explanation:**

This paper introduces a framework for continual learning that accounts for concept drift, where the distributions of previously learned classes can change over time. The authors create four new benchmark datasets by applying transformations to existing vision datasets to simulate this drift.  All the reviewers believed the claims are generally supported by the empirical evidence with some aspects to be improved. After the discussion, they have no substantial concerns.